# The role of electrical conductivity in radar wave reflection from glacier beds

Slawek M. Tulaczyk[1], Neil T. Foley[1]

[1]Department of Earth and Planetary Sciences, University of California, Santa Cruz, CA 95064, USA

5 *Correspondence to*: Slawek M. Tulaczyk (stulaczy@ucsc.edu)

**Abstract.** We have examined a general expression giving the specular reflection coefficient for a radar wave approaching a reflecting interface with normal incidence. The reflecting interface separates two homogeneous isotropic media, the properties of which are fully described by three scalar quantities: dielectric permittivity, magnetic permeability, and electrical conductivity. The derived relationship

10  indicates that electrical conductivity should not be neglected *a priori* in glaciological investigations of subglacial materials, and in GPR studies of saturated sediments and bedrock, even at the high end of typical linear radar frequencies used in such investigations (e.g., 100-400 MHz). Our own experience in resistivity surveying in Antarctica, combined with a literature review, suggests that a wide range of geologic materials can have electrical conductivity that is high enough to significantly impact the value

15  of radar reflectivity. Furthermore, we have given two examples of prior studies in which inclusion of electrical conductivity in calculation of the radar bed reflectivity may provide an explanation for results that may be considered surprising if the impact of electrical conductivity on radar reflection is neglected. The commonly made assumption that only dielectric permittivity of the two media need to be considered in interpretation of radar reflectivity can lead to erroneous conclusions.

 **1 Introduction**

Ice penetrating radar represents the most successful geophysical technique in glaciology, that efficiently yields observational constraints on fundamental properties of land ice masses on Earth, such as thickness, internal structures, and bed properties (e.g., reviews in Plewes and Hubbard, 2001; Dowdeswell and Evans, 2004). Radar has also been used to investigate ice masses on Mars (e.g., Holt et al., 2008; Bierson et al., 2016) and will be used to probe ice shells on icy satellites (e.g., Chyba et al., 1998; Aglyamov et al., 2017). Much of the success of radar imaging in glaciology can be attributed to the fact that glacier ice is a polycrystalline solid with either no, or little, liquid water and low concentration of impurities from atmospheric deposition, e.g., sea salts and acidic impurities (Stillman et al., 2013). Hence, glacier ice is a poor electrical conductor and is quite transparent to electromagnetic waves over a broad range of frequencies (Dowdeswell and Evans, 2004). Radar systems used for deep ice imaging have generally evolved over the last several decades from low-frequency radars (1-10 MHz; e.g., Catania et al., 2003) towards systems which can penetrate kilometers of ice at frequencies reaching above 100 MHz (e.g., Winter et al., 2017).

Electrical conductivity is the material property that controls attenuation of electromagnetic waves (Stratton, 1941) and the resistive nature of glacier ice makes it reasonable to assume that it is a nearly lossless material with regards to radar wave propagation. However, as illustrated by the research on the origin of internal radar reflectors in ice sheets and glaciers, radar reflections can be caused by contrasts in either real permittivity or conductivity, even though such englacial contrasts are quite small for both of these material properties (Paren and Robin, 1975). These authors developed two different equations

for the radar reflection coefficient, which express the dependence of this coefficient on, separately, permittivity and conductivity contrasts (*ibid.*, p. 252). This is a common approach to get around the fact that the full version of the radar reflection coefficient involves complex quantities (Dowdeswell and Evans, 2004, eq. 7; Bradford, 2007). Whereas radar waves can typically transmit much energy through

weak englacial reflectors and provide information on the structure over a large range of ice thicknesses, the radar reflectivity of the ice bed offers basically the only insight from radar surveys into the nature of geologic materials underlying ice masses. This is because sub-ice environments are typically not imaged directly by ice penetrating radars (Plewes and Hubbard, 2001). Rather, inferences about sub-ice conditions, e.g., the presence or absence of subglacial water, are drawn from the lateral and temporal

variations in radar bed reflectivity (e.g., Catania et al., 2003; Chu et al., 2016).

Here, we build on the pioneering work of Stratton (1941) to propose a version of the specular radar amplitude reflection coefficient, which retains both real permittivity and conductivity of the two media that are separated by the reflecting interface. The advantage of this approach over past studies treating

the impact of electrical conductivity on radar reflectivity (e.g., Peters et al., 2005; MacGregor et al., 2011; Christianson et al., 2016) is that the reflectivity equations presented here do not use complex variables. Furthermore, we overview constraints on the electrical conductivity of plausible subglacial materials and illustrate how consideration of the impact of electrical conductivity on radar bed reflection can improve glaciological interpretations of subglacial conditions.

In general, the mathematical treatment of propagation and reflection of electromagnetic (henceforth

EM) waves includes three fundamental properties of the media through which EM waves propagate:

dielectric permittivity, $\varepsilon$; electric conductivity, $\sigma$; and magnetic permeability, $\mu$. Maxwell's equations

for EM waves in homogeneous and isotropic media illustrate the role of these properties in EM wave

propagation (Stratton, 1941, p. 268):

$$\nabla \times \boldsymbol{E} + \mu \frac{\partial \boldsymbol{H}}{\partial t} = 0 \tag{1a}$$

$$\nabla \times \boldsymbol{H} - \varepsilon \frac{\partial \boldsymbol{E}}{\partial t} - \sigma \boldsymbol{E} = 0 \tag{1b}$$

$$\nabla \cdot \boldsymbol{H} = 0 \tag{1c}$$

$$\nabla \cdot \boldsymbol{E} = 0 \tag{1d}$$

where $\boldsymbol{E}$ denotes the electric field intensity vector, $\boldsymbol{H}$ is the magnetic field intensity vector, and $t$ is time.

Magnetic permeability and dielectric permittivity are associated with time derivatives of the magnetic

and electric field intensities, respectively (Eq. 1ab). Their values are never zero, even in free space, and

they can be thought of as an analog for elastic constants used in description of seismic wave

propagation. The free space values of $\varepsilon_o = 8.8541878128 \times 10^{-12}$ s$^2$H$^{-1}$m$^{-1}$ and $\mu_o = 1.25663706212 \times$

$10^{-6}$ H m$^{-1}$ are used in physics and geophysics as reference quantities, so that, for instance, relative

dielectric permittivity, sometimes also referred to as the specific inductive capacity, is defined as $\varepsilon_r =$

$\varepsilon/\varepsilon_o$. In contrast to magnetic permeability and dielectric permittivity, electric conductivity can be zero

(e.g., free space) or negligibly small (e.g., glacier ice). In such media, EM waves can propagate (nearly)

without loss of amplitude since conductive electric currents, represented in Eq. (1b) by the third term on

the left-hand side, provide the physical mechanism for EM wave attenuation. It is worth noting that in

geophysical literature it is often customary to substitute electrical resistivity, $\rho$, expressed in $\Omega$m, for

electrical conductivity, $\sigma$, with units of S m$^{-1}$. It is straightforward to switch between the two since one

is simply the reciprocal of the other, such that $\rho = 1/\sigma$, or vice versa. Another noteworthy fact is that

most materials on and near the Earth's surface, including most common minerals, rocks, ice, and water,

have magnetic permeability that is not significantly different from that of free space, $\mu_o$, except for a

small subset of minerals that are not very abundant (O'Reilly, 1976; Keller, 1988). Later this will

become important because it will enable us to eliminate magnetic permeability from the equations

describing radar wave reflection, in which it appears both in the numerator and denominator. This will

simplify the problem of radar reflection to a function of just two material properties: electrical

conductivity and dielectric permittivity.

Before focusing on analyses of EM wave reflection, we note that Stratton (1941, section 5.2) proposed

solutions describing propagation and reflection of harmonic plane waves in the homogeneous and

isotropic media by using a complex propagation constant, $\mathbf{k}$, defined as (*ibid,* eq. 30):

$$\mathbf{k} = \alpha + i\beta \tag{2}$$

where $\alpha$ is the phase constant and $\beta$ is the attenuation factor while $i$ is the standard imaginary unit, such

that $i^2 = -1$. We note that throughout this paper, we will use bold type for symbols designating complex

quantities. The complex propagation constant plays a crucial role in Stratton's expressions for the

reflection coefficient. It should be noted that in geophysical literature, the meaning of symbols $\alpha$ and $\beta$

is sometimes switched, so that the former is the attenuation factor (e.g., Knight, 2001, p. 231). Since

Stratton's work provides the basis for our analyses, we will keep using his terminology here. The two components of the propagation constant are given by (Stratton, 1941, eqs. 48 and 49):

$$\alpha = \omega \left[ \frac{\mu\varepsilon}{2} \left( \sqrt{1 + \frac{\sigma^2}{\varepsilon^2\omega^2}} + 1 \right) \right]^{1/2} = \omega \left[ \frac{\mu\varepsilon}{2} \left( \sqrt{1 + \psi^2} + 1 \right) \right]^{1/2} \tag{3a}$$

$$\beta = \omega \left[ \frac{\mu\varepsilon}{2} \left( \sqrt{1 + \frac{\sigma^2}{\varepsilon^2\omega^2}} - 1 \right) \right]^{1/2} = \omega \left[ \frac{\mu\varepsilon}{2} \left( \sqrt{1 + \psi^2} - 1 \right) \right]^{1/2} \tag{3b}$$

where $\omega$ is the angular frequency, related to the linear frequency $f$ through $\omega = 2\pi f$, and all other symbols have been already defined. For use in subsequent discussions we have defined a control parameter $\psi = \sigma/(\varepsilon\omega)$ whose physical meaning is analyzed in the next paragraph. It is of paramount importance to our later analyses to note after Stratton (1941, p. 276) "…that $\alpha$ and $\beta$ must be real." Hence, the only imaginary part of the complex propagation constant, $\boldsymbol{k}$, is due to the term $\boldsymbol{i\beta}$ on the

right-hand side of Eq. (2). Although the material properties such as electrical permittivity and conductivity may themselves be expressed as complex quantities (e.g., Bradford, 2007), Eq. (3ab) require real values of all three material parameters, $\varepsilon$, $\sigma$, $\mu$, applicable at a specific angular frequency, $\omega$ (Stratton, 1941, p. 511).

Our subsequent discussion of Equations (3ab) will reveal three general modes of behavior of the propagation constant that are governed by the value of the control parameter $\psi = \sigma/(\varepsilon\omega)$, which is related to the ratio of half of the wavelength in a non-conductive material, $\lambda/2 = \pi/(\omega\sqrt{\varepsilon\mu})$, to the conductive skin depth, $\delta = \sqrt{2/(\omega\mu\sigma)}$ (Stratton, 1941, eq. 66). These two length scales are important in the context of electromagnetic wave reflection (Figure 1A). When the medium underlying the

reflecting interface is a non-conductive dielectric, it needs to have a thickness of at least $\lambda/2$ for its

properties to fully determine the reflection strength (e.g., Church et al., 2020, figure 9). So, a radar wave

reflecting from an interface between two perfect dielectric materials is sensitive to the properties of the

sub-interface material to within about $\lambda/2$ below the interface. The skin depth, in turn, reflects the e-

folding length scale to which the reflecting wave induces electric eddy currents in the sub-interface

medium (Stratton, 1941, p. 504). The ratio of the two length scales is (to within a factor of $\pi/4$) given by

$\sqrt{\sigma/(\omega\varepsilon)} = \sqrt{\psi}$, and its fourth power controls the relative importance of electrical conductivity in

Equations 3ab. When the deeper material is conductive, $\delta$ is much shorter than $\lambda/2$ and when its

conductivity is low, the opposite is true. Hence, the ratio of $\lambda/2$ to $\delta$ can be used as a gauge of the

relative importance of displacement and conduction currents in the process of wave reflection.


The simplest version of Equations 3ab is obtained when electrical conductivity is either zero or

negligible ($\sigma \ll \varepsilon\omega$ or $\psi \ll 1$) so that the phase and attenuation factors simplify to:

$$\alpha = \omega\sqrt{\mu\varepsilon} \tag{4a}$$

$$\beta = 0 \tag{4b}$$

and the propagation constant, which is no longer a complex quantity since $\beta = 0$, becomes:

$$k = \alpha = \omega\sqrt{\mu\varepsilon} \tag{4c}$$

This assumption is often made in glaciological and geophysical radar interpretation (e.g., Knight, 2001;

Plewes and Hubbard, 2001; Dowdeswell and Evans, 2004) and it is certainly justified for glacier ice,

which has sufficiently low conductivity at a wide range of frequencies (e.g., Stillman et al., 2013).

Glacier ice, and other materials for which $\psi \ll 1$, can be classified as good dielectrics with low loss with respect to propagation of EM waves (Figure 1B). At the opposite end of the spectrum, when $\psi \gg 1$, the material can be classified as high-loss, poor dielectric medium (Figure 1B) and Eq. (3ab) simplify to:

$$\alpha = \beta = \sqrt{\frac{\mu\omega\sigma}{2}} \tag{5a}$$

and the complex propagation constant becomes:

$$\boldsymbol{k} = \alpha(1 + \boldsymbol{i}) = \beta(1 + \boldsymbol{i}) = \sqrt{\frac{\mu\omega\sigma}{2}}(1 + \boldsymbol{i}) \tag{5b}$$

The full versions of Eq. (2) and (3ab) are, thus, only needed when dealing with the transitional region corresponding approximately to conditions when $0.1 < \psi < 10$. In Figure 1B, these limits correspond to ca. 5-10% error in the low-loss and high-loss values of $\alpha$ and $\beta$, Eq. (4ab) and (5a), as compared to

their values calculated using Eq. (3ab). In practical applications of radar reflectivity investigations, the challenge, of course, is that it may be impossible to know *a priori* what the electrical conductivity of the target material is and to decide which form of the propagation constants is applicable.

### 3 The Low-Loss Assumption and Its Limitations

As can be easily gleaned from Eq. (4abc), the most convenient simplification of Eq. (2) and (3ab) results from the low-loss assumption, $\sigma \ll \varepsilon\omega$, ($\psi \ll 1$) because the propagation constant is then no longer a complex number and one material property, electrical conductivity, can be completely eliminated from further consideration. As mentioned above, this assumption is a reasonable one for

glacier ice. However, it cannot be necessarily assumed to generally hold for subglacial materials such as

saturated bedrock and sediments or for marine accreted ice of ice shelves.

Figure 1C allows us to verify if the range of electrical conductivity and relative permittivity for common geologic materials justifies the low-loss assumption. For illustration purposes, we use three different linear frequencies, $f$, of 1, 10, and 100 MHz, which are representative of the range of linear

frequencies used in glaciology, planetary science, and ground penetrating radar (GPR) investigations (e.g., Jacobel and Raymond, 1984; Catania et al., 2003; Bradford, 2007; Holt et al., 2008; Mouginot et al., 2014). As a reminder, the angular frequency is related to the linear frequency by: $\omega = 2\pi f$. The relative permittivity considered in Figure 1C spans that expected for common minerals and rocks in dry conditions at the low end to 100% liquid water by volume at the high end (Midi et al., 2014; Josh and

Clennell, 2015). For each of the considered frequencies, the range of electrical conductivities for which neither the low-loss, nor the high-loss, assumption is truly justified covers about one order of magnitude. The exact conductivity values falling within this range are dependent on relative permittivity. For instance, for 100 MHz linear frequency, the low-loss limit corresponds to conductivity of ca. 0.01 S m$^{-1}$ (resistivity of ca. 100 Ωm) for $\varepsilon_r = 5$, typical for dry minerals and rocks (e.g, Josh and

Clennell, 2015), but is an order of magnitude higher ($\sigma = 0.1$ S m$^{-1}$ and $\rho = 10$ Ωm) for $\varepsilon_r = 55$, which would be expected either for clay-poor sediments with very high water content or saturated clay-rich sediments (Arcone et al., 2008; Josh and Clennell, 2015).

Most common minerals have by themselves negligibly small electrical conductivity at pressures and

temperatures prevailing near the surface of the Earth, except for metallic minerals and minerals

exhibiting semiconductive behavior, like sulfides, oxides, and graphite (e.g., Keller, 1998). As

embodied in the empirical Archie's law, the bulk electrical conductivity of sediments and rocks is

mainly due to electrolytic conduction associated with the presence of liquid water and solutes in pore

spaces and fractures (Archie, 1942). When re-written in terms of electrical conductivity, the original

Archie's relation (Archie, 1942, eq. 3) becomes:

$$\sigma = \sigma_w \phi^m \tag{6}$$

where $\sigma_w$ is the conductivity of pore fluid, $\phi$ is the porosity, expressed as a volume fraction of pore

spaces, and $m$ is the empirical cementation exponent. This relationship was originally developed for

clean sandstone and is less applicable to fine-grained, particularly clay-bearing, rocks and sediments for

which surface conduction becomes important (Ruffet et al., 1995). This long-known conductive effect

(Smoluchowski, 1918), represents an enhancement of electrolytic conduction near charged solid

surfaces and its magnitude tends to scale with the specific surface area of sediments (e.g., Arcone et al.,

2008; Josh and Clennel, 2015).

Overall, the low-loss assumption is less likely to be applicable in three general types of geologic

materials: (1) ones containing sufficient concentration of conductive minerals (e.g., Hammond and

Sprenke, 1981), (2) sediments and rocks saturated with high conductivity fluids, and (3) saturated clay-

bearing rocks and sediments. If we take the low-loss conductivity limits for 100 MHz frequency from

Figure 1C, 0.01-0.1 S m$^{-1}$, and apply them to the compilation of electrical conductivity for geologic

materials in figure 1 of Ruffet et al. (1995) the low-loss assumption is questionable for a wide range of

materials, including shales, sandstones, coal, metamorphic rocks, igneous rocks as well as graphite and

sulfides. This simplifying assumption is even more generally suspect for lower frequencies, such as 1

and 10 MHz in Figure 1C.

The compilation data in figure 1 of Ruffet et al. (1995) can be criticized as being overly generalized and

we turn now to some specific relevant studies. In our regional helicopter-borne time-domain EM survey

of liquid-bearing subglacial and sub-permafrost materials performed in McMurdo Dry Valley region in

Antarctica we mostly observed electrical resistivities of 1-100 $\Omega$m ($\sigma$ = 0.01-1 S m$^{-1}$) (Dugan et al.,

2015; Mikucki et al., 2015; Foley et al., 2016; Foley et al., 2019ab). Extensive regional DC and EM

surveys of Pleistocene glacial sequences in Denmark and Germany yielded resistivities in the same

range of values, except for clean outwash sand and gravel which tend to be more resistive (Steuer et al.,

2009; Jørgensen et al., 2012). Hence, these results of regional resistivity surveys in modern and past

glacial environments also support the contention that the low-loss assumption is not generally

applicable to geologic materials expected beneath glaciers and ice sheets, or in post-glacial landscapes.

Although our focus here is on glacial environments, we conjecture based on our review of available

constraints that it may be similarly problematic to make such blanket low-loss assumption in GPR

investigations of reflectors in other saturated sediments (e.g., Bradford, 2007).

The table below summarizes values of relative permittivity and electrical conductivity for materials that

can be found at the base or beneath ice sheets and glaciers (Table 1). These values come from a

combination of sources, including past compilations (e.g., Peters et al., 2005, table 1 and Christianson et al., 2016, table 1) as well as laboratory and field measurements. Whereas the laboratory measurements were typically conducted at radar frequencies, most field measurements of conductivity of glacial materials come from Airborne ElectroMagnetics (AEM) surveys over formerly glaciated regions in

Europe and North America. The AEM sensors operate typically in frequency ranges <1 MHz. For instance, the AEM sensor used by us in Antarctica is a broadband time-domain AEM sensor covering frequencies from 1 Hz to 300 kHz (e.g., Foley et al., 2016). The three columns on the right side of Table 1 give the corresponding amplitude reflection coefficients calculated using equations derived and discussed in the next section.


**Table 1.** We compile relative permittivity, $\varepsilon_r$, and electrical conductivity, $\sigma$, for glacier ice and likely basal and subglacial materials. Whenever possible, the values are reported for temperatures close to the freezing point of freshwater and linear frequencies of 10s to 100s of MHz. The permittivity and conductivity values are followed by the corresponding dimensionless control parameter $\psi = \sigma/(\varepsilon\omega)$ for 10 and 100 MHz. For each basal and subglacial material, we also give the

values of the amplitude reflection coefficient, $r$ (Eq. 7b), and the power reflection coefficient, $R$ (Eq. 9), for a specular basal interface at frequencies of 10 and 100 MHz. $R$ is in decibels, and $r$ is in percent. The last column gives the absolute value of the frequency-independent $r$ under the assumption of zero conductivity (Eq. 11).

| Material | $\varepsilon_r$ | $\sigma$ S/m | $\psi_{10MHz}$; $\psi_{100MHz}$ ND; ND | $r_{10MHz}$; $R_{10MHz}$ %; dB | $r_{100MHz}$; $R_{100MHz}$ %; dB | $\|r_{\sigma=0}\|$ % |
|---|---|---|---|---|---|---|
| Glacier ice | 3.2[a] | 0.00007[a] | 0; 0 | | | |
| Frozen bedrock | 2.7[a] | 0.0002[a] | 1; 0 | 5; -26.6 | 4; -27.5 | 4 |
| Marine ice | 3.4[a] | 0.0003[b] | 10; 1 | 4; -29.1 | 2; -36.5 | 2 |
| Saturated bedrock | 4-15[c] | 0.001-0.01[d] | 1-28; 0-3 | 12-50; -18.1 to -6.0 | 6-37; -24.9 to -8.6 | 6-37 |
| Saline basal ice | 3.4[a] | <0.02[e] | <66; <7 | <65; <-3.8 | 23; <-12.7 | 2 |
| Sandy till | 6-20[a] | <0.02[f] | <11-38; <1-4 | <62-64; <-4.2 to -3.9 | <24-43; <-12.7 to -7.3 | 16-43 |
| Subglacial water | 88[g] | 0.04[h] | 5; 1 | 73; -2.8 | 68; -3.3 | 68 |
| Fairbanks silt | 24[i] | 0.043[i] | 20; 2 | 72; -2.8 | 48; -6.4 | 47 |
| Clay-bearing till | 6-20[a] | 0.015-0.1[j] | 8-188; 1-19 | 59-81; -4.6 to -1.8 | 21-52; -13.7 to -5.7 | 16-43 |
| Clay | 31[k] | 0.24[k] | 36; 4 | 88; -1.1 | 65; -3.8 | 51 |
| Marine clay | 31[l] | 0.1-1[m] | 36-364; 4-36 | 81-94; -1.8 to -0.5 | 54-82; -5.3 to -1.8 | 51 |

| *Seawater* | 79[n] | 2.9[o] | 415; 41 | 97; *-0.3* | 89; *-1.0* | 67 |
| *Brine* | 62[p] | 4.8[q] | 874; 87 | 97; *-0.2* | 92; *-0.8* | 63 |

[a] Christianson et al. (2016, table 1); [b] Conductivity measured at 150 MHz on ice samples from the Westphal Ice Shelf (Moore et al., 1994, figure 6); [c] Various bedrock lithologies from Davis and Annan (1989, table 1); [d] Approximate spread of median values for various bedrock lithologies as measured using an AEM sensor spanning frequency 0.9 kHz to 25 kHz (White and Beamish, 2014, table 2); [e] Estimated from figure 6 in Moore et al. (1994) using the maximum salinity (15 ppt) of basal ice samples from Taylor Glacier, Antarctica (Montross et al., 2014, figures 2 and 4); [f] Schamper et al. (2014, table 1); [g] Value of 86 measured at 200 MHz and temperature 5ºC but temperature-corrected by us to 88 based on Buchner et al. (1999, figure 2); [h] Water conductivity measured in Subglacial Lake Whillans of 0.072 S m$^{-1}$ reported for temperature of 25ºC (Christner et al. 2014, table 1) and corrected to 0ºC (Hayashi, 2004); [i] Value for a sediment sample with 39% porosity of which three quarters were saturated with deionized water (Arcone et al., 2008, figure 8 for 100 MHz); [j] AEM surveys of glacial sequences in Schamper et al. (2014, table 1), Høyer et al. (2015, figures 5 and 6), Jørgensen et al. (2015, figure 2); [k] Value for clay fraction with 56% porosity of which 60% were saturated with deionized water (Arcone et al., 2008, figure 8 for 100 MHz); [l] Assuming the same value as for the clay fraction from Arcone et al. (2008); [m] The high bound is from table 1 in Schamper et al. (2014) with other values from Burschil et al. (2012) and Høyer et al. (2015); [n] Seawater value of 77 measured at 5ºC and temperature corrected by us to 79 (Buchner et al., 1999, figure 2); [o] Mikucki et al. (2015, table 1); [p] Used the salinity of Blood Falls brine from Lyons et al. (2019) to arrive at this estimate for 0ºC using figure 2 in Buchner et al. (1999); [q] West Lake Bonney brine from Mikucki et al. (2015, table 1).

## 4 General and Simplified Forms of the Radar Reflection Coefficient

In order to illustrate the general form of the radar reflection coefficient we start with the expression derived by Stratton (1941, chapter 9) for a reflecting interface separating two homogeneous and isotropic half spaces characterized by three scalar material properties each: $\varepsilon_1$, $\varepsilon_2$, $\sigma_1$, $\sigma_2$, $\mu_1$, $\mu_2$ (Figure 1A). We limit ourselves to considering specular reflection of a plane wave approaching the interface at normal incidence from medium 1 towards medium 2 (adapted from Stratton, 1941, p. 512, eq. 11):

$$r \equiv \frac{E_r}{E_o} = \frac{\mu_2 k_1 - \mu_1 k_2}{\mu_2 k_1 + \mu_1 k_2} \qquad\qquad (7a)$$

where $r$ is the complex reflection coefficient, defined as the complex intensity of the reflected wave, $E_r$, normalized by the complex intensity of the incident wave, $E_o$. The materials on both sides of the reflecting interface are characterized by complex propagation constants, $k_1$ and $k_2$, which are related to

the respective material constants characterizing the media (i.e., $\varepsilon_1$, $\varepsilon_2$, $\sigma_1$, $\sigma_2$, $\mu_1$, $\mu_2$) through Eq. (2) and (3ab) (Figure 1A).

From this point going forward in our analysis we will assume that both of the media have the magnetic

permeability of free space, as it is reasonable to do for most rocks and minerals at temperatures and

pressures near the surface of the Earth. With this simplification Eq. (7a) becomes:

$$r = \frac{k_1 - k_2}{k_1 + k_2} = \frac{\alpha_1 + i\beta_1 - \alpha_2 - i\beta_2}{\alpha_1 + i\beta_1 + \alpha_2 + i\beta_2} = \frac{(\alpha_1 - \alpha_2) + i(\beta_1 - \beta_2)}{(\alpha_1 + \alpha_2) + i(\beta_1 + \beta_2)} \tag{7b}$$

where we have expanded the right-hand side of this equation using the complex propagation constants,

$k_1$ and $k_2$, (Eq. 2) for both media. The real amplitude reflection coefficient, $r$, can be expressed as the

absolute value of the complex vector $r$:

$$r = |r| = \sqrt{\frac{(\alpha_1 - \alpha_2)^2 + (\beta_1 - \beta_2)^2}{(\alpha_1 + \alpha_2)^2 + (\beta_1 + \beta_2)^2}} \tag{8}$$

where the absolute value is, by definition, the Pythagorean length of the complex vector, $r$, in the

complex plane (Argand Diagram).

The power reflection coefficient, $R$, is the square of Eq. (8) (Stratton, 1941, p. 512, eq. 12):

$$R = \frac{(\alpha_1 - \alpha_2)^2 + (\beta_1 - \beta_2)^2}{(\alpha_1 + \alpha_2)^2 + (\beta_1 + \beta_2)^2} \tag{9}$$

It is worth noting that Eq. (8) and (9) are, on their own, underconstrained. At least in glaciology, one

can put reasonable constraints on the electrical conductivity and permittivity of ice, $\sigma_1$ and $\varepsilon_1$ (e.g.,

Stillman et al., 2013) (Table 1), which, in this example, corresponds to the medium 1 through which the

incident wave is propagating towards the reflecting interface (Figure 1A). The two unknowns are then

the electrical conductivity and permittivity, $\sigma_2$ and $\varepsilon_2$, of the medium underlying ice (Table 1).

Additional constraint can be gained from the tangent of the phase shift angle of the reflected wave, given by (Stratton, 1941, p. 513, eq. 15):

$$\tan(\varphi) = \frac{2(\alpha_2\beta_1 - \alpha_1\beta_2)}{(\alpha_1{}^2 + \beta_1{}^2) - (\alpha_2{}^2 + \beta_2{}^2)} \tag{10}$$

So, if radar reflectivity and phase shift, $\varphi$, can be measured accurately enough then, at least in principle,

Eq. (8) and (10) represent a system of two equations with two unknowns, $\sigma_2$ and $\varepsilon_2$. However, we will

later illustrate the limitations of this approach that are related to the fact that in both limiting regimes,

the low-loss and the high-loss one, the tangent of the phase shift angle is small.

Let us now examine the two limiting cases of Eq. (9), first when the sub-ice material is low loss and

then when it is high loss. In the first case, $\sigma_2 \ll \varepsilon_2\omega$ ($\psi \ll 1$), we substitute Eq. (4ab) for $\alpha_1$, $\alpha_2$, and $\beta_1$,

$\beta_2$ in Eq. (8) and obtain:

$$r = \sqrt{\frac{(\alpha_1 - \alpha_2)^2}{(\alpha_1 + \alpha_2)^2}} = \frac{\alpha_1 - \alpha_2}{\alpha_1 + \alpha_2} = \frac{\sqrt{\varepsilon_1} - \sqrt{\varepsilon_2}}{\sqrt{\varepsilon_1} + \sqrt{\varepsilon_2}} \tag{11}$$

The reflection coefficient simplifies to a function of only permittivities of ice, $\varepsilon_1$, and the sub-ice

geologic material, $\varepsilon_2$. This is an encouraging result because it agrees with a widely used form of radar

reflection coefficient in the case of an interface between two perfect dielectrics (e.g., Knight, 2001). The

tangent of the phase shift angle (Eq. 10) is always zero for the low-loss case but the phase shift angle is

either zero, when $r$ values are positive, or 180° when they are negative.

For the second case, we assume that ice (medium 1 in Figure 1A) is still a lossless dielectric but that the

sub-ice medium is high loss, $\sigma_2 \gg \varepsilon_2\omega$ ($\psi \gg 1$), so that we use Eq. (4ab) for $\alpha_1$, $\beta_1$, and Eq. (5a) for $\alpha_2$,

$\beta_2$ in Eq. (8):

$$r = \sqrt{\frac{(\alpha_1-\alpha_2)^2+\beta_2{}^2}{(\alpha_1+\alpha_2)^2+\beta_2{}^2}} = \sqrt{\frac{\omega\varepsilon_1-\sqrt{2\varepsilon_1\omega\sigma_2}+\sigma_2}{\omega\varepsilon_1+\sqrt{2\varepsilon_1\omega\sigma_2}+\sigma_2}} \approx \sqrt{\frac{\sigma_2-\sqrt{2\varepsilon_1\omega\sigma_2}}{\sigma_2+\sqrt{2\varepsilon_1\omega\sigma_2}}} \qquad (12)$$

where the final, approximate, expression on the right-hand side is taking advantage of the fact that,

under the high-loss assumption, $\sigma_2 \gg \varepsilon_1\omega$ ($\psi \gg 1$) given that the permittivity of ice is low (Stillman et

al., 2013). As shown by Eq. (12), the high-loss version of the reflection coefficient is sensitive to the

angular frequency, $\omega$, the permittivity of ice, $\varepsilon_1$, and electrical conductivity of the sub-ice material, $\sigma_2$.

Since the radar frequency and the permittivity of ice are known, Eq. (12) can be re-arranged to calculate

the subglacial electrical conductivity from radar reflection strength, if one assumes the high-loss case:

$$\sigma_2 \approx \frac{2\varepsilon_1\omega(r^2+1)^2}{(r^2-1)^2} = \frac{2\varepsilon_1\omega(R+1)^2}{(R-1)^2} \qquad (13)$$

where all the symbols have been defined previously. This approach is a counterpart to the common

practice of using Eq. (11) to calculate permittivity of the sub-ice material under the low-loss

assumption.

**5 Discussion**

Figure 1D shows the full version of the amplitude reflection coefficient (Eq. 8) plotted for the case of

100 MHz linear frequency and a range of relative permittivities (in this case $\varepsilon_r = \varepsilon_2/\varepsilon_o$) and electrical

conductivities for the sub-ice material. The family of horizontal line segments on the left corresponds to

the case of lossless dielectric media being present beneath ice. These line segments can be

approximated by Eq. (11), which is commonly used in glaciology and GPR studies to make inferences about the nature of geologic materials. Due to the fact that common minerals have relatively low relative permittivity (4-10) and liquid water has very high relative permittivity (Midi et al., 2014), the strength of the basal reflection coefficient is often interpreted solely as the function of water content. This is also a common practice in GPR investigations of interfaces between sediment layers (e.g., Stoffregen et al., 2002). In glaciology and planetary science, for instance, bright radar reflectors have been used in the search for subglacial lakes on Earth and Mars because open water bodies beneath ice should be the most reflective subglacial materials, at least in the low loss regime described by Eq. (11) (Plewes and Hubbard, 2001; Dowdeswell and Evans, 2004; Orosei et al., 2018).

Starting at electrical conductivities of about 0.01-0.1 S m$^{-1}$ (resistivity of 10-100 $\Omega$m), the reflection coefficient for 100MHz frequency becomes increasingly more dependent on the conductivity than on the permittivity of the sub-ice material. At conductivities greater than 0.1 S m$^{-1}$ (resistivity of 10 $\Omega$m), the coefficient is for all practical purposes independent of relative permittivity of subglacial materials and rises in value above its high value of 0.68 characterizing the ice-above-water scenario under lossless conditions (Table 1). This means that high conductivity subglacial materials can appear at least as bright as subglacial lakes filled with fresh meltwater. Such high conductivity materials can include seawater- or brine-saturated sediments and bedrock (Foley et al., 2016, table 2) as well as clay-bearing sediments or bedrock saturated with natural waters of any reasonably high conductivity (Table 1). Large parts of the Antarctic ice sheet are underlain by clay-rich subglacial tills, which may contain over 30% clay (Tulaczyk et al., 1998; Studinger et al., 2001; Tulaczyk et al., 2014; Hodson et al., 2016).

Relatively high scattering from a rough interface between ice and clay-bearing, reflective bedrock may keep radioglaciologists from interpreting such a setting as a subglacial lake. But clay-bearing subglacial sediments may also have very low shear strength (e.g., Tulaczyk et al., 2001) resulting in an ice-sediment interface that has low roughness over length scales comparable to radar wavelengths (e.g., ca.

1 m for 100 MHz radar) and may not be distinguishable from an ice-water interface on the basis of scattering or reflectivity.

The effect of electrical conductivity of subglacial materials on basal radar reflectivity may be responsible for some past puzzling glaciological radar results. For instance, Christianson et al. (2012)

used a 5 MHz center frequency radar to perform extensive mapping of basal reflectivity around Subglacial Lake Whillans. They failed to find a relationship between the outline of the lake inferred from satellite altimetry and the observed pattern of basal radar reflectivity. Subsequent drilling found very clay-rich sediments in the region (Tulaczyk et al., 2014; Hodson et al., 2016) and such subglacial sediments can be conductive enough to produce radar reflectivity that is the same, or higher, than

reflectivity from an ice-lake interface (e.g., Arcone et al., 2008). This is particularly the case for low frequency radar waves with center frequency of 5MHz, for which only subglacial materials that are less conductive than ca. 0.01-0.001 S m$^{-1}$ (resistivity of 100-1000 $\Omega$m), depending on permittivity, will meet the criterion of a low-loss material. Moreover, high-porosity, fine-grained subglacial sediments are also likely to be deformable and will make for a relatively smooth ice-bed contact, which is sometimes used

as an additional criterion in mapping of ponded subglacial waters (e.g., Oswald et al., 2018). Hence,

areas of clay-rich subglacial sediments surrounded by bedrock may be misinterpreted as areas of subglacial water ponding.

In the same general part of Antarctica, MacGregor et al. (2011) mapped basal reflectivity across the grounding zone of Whillans Ice Stream using a 2 MHz radar. Their survey found no clear increase in radar reflectivity across the grounding line, where the ice base goes from being underlain by saturated sediments to floating on seawater. If one interprets this setting in the context of the low-loss assumption (Eq. 11), basal reflectivity should be higher over seawater than sediments (Arcone et al., 2008; Midi et al., 2014). However, Eq. (12) solved for a 2MHz linear frequency (detailed results not shown here) shows a high reflection coefficient of ca. 0.9 for all subglacial materials with conductivity higher than 0.05 S m$^{-1}$ (resistivity of 20 $\Omega$m). Since seawater has electrical conductivity of ca. 2.9 S m$^{-1}$ (0.35 $\Omega$m) and the clay rich subglacial sediments in the region can have conductivity >0.05 S m$^{-1}$ (<20 $\Omega$m) (Table 1), the radar survey of MacGregor et al. (2011) may have encountered a problem arising from the high-loss end member of the reflection coefficient (Eq. 12). In this regime, the reflection coefficient is no longer sensitive to relative permittivity so that transition from saturated sediments to pure water no longer increases the reflection coefficient. At the same time, the value of reflectivity calculated from Eq. (12) changes only slightly with changes in already high electrical conductivity so that differences in conductivity between seawater and clay-rich sediments may be too small to be detectable in noisy radar reflection data, particularly if the sediments themselves are saturated by seawater or brackish porewater (e.g., marine clay in Table 1). In general, grounding zones may prove to be one of the most important subglacial environments in which radioglaciologists have to consider the electrical conductivity of

subglacial materials, in addition to their permittivity. In this environment, one is reasonably likely to

encounter both clay-rich sediments and high-conductivity fluids. For instance, high bed reflectivity

observed on the upstream side of a grounding zone may be interpreted as a sign of seawater intrusion

but it may as well be caused by clay-rich marine sediments that are now being overridden by the ice

base (Table 1).

It is beyond the scope of this manuscript to analyze and critique specifics of the multitudes of relevant

radioglaciological studies. Our goal is to argue that, in some circumstances, radar bed reflectivity can be

a function of subglacial clay content and water salinity, rather than being just purely determined by bed

water content, through its impact on bed permittivity (Table 1). The latter line of reasoning is present in

the radioglaciological literature (e.g., Oswald and Gogineni, 2008), although it should be noted that in

this specific study the use of high center linear frequency (150 MHz) may help diminish the effects of

subglacial electrical conductivity on bed reflectivity (Table 1). Another example of radioglaciological

application in which one should carefully consider the potential impact of electrical conductivity on bed

reflectivity is mapping of frozen and melted bed zones (e.g., Chu et al., 2018). In this case, a reflectivity

contrast between water-saturated, low-porosity, low-conductivity bedrock (e.g., $r = 0.057$ for 100 MHz

in Table 1) and zones of subglacial clay-bearing till (e.g., $r = 0.519$ for 100 MHz in Table 1) may reach

about 20 dB in terms of power reflectivity contrast. Such large contrast could be interpreted as a

transition from frozen to melted bed despite the fact that both materials may contain liquid water in

reality. Radar mapping of zones of basal freezing could be further confounded by the fact that basal

freezing can lead to cryoconcentration of solutes in the remaining subglacial liquid water (e.g., Foley et

al., 2019b). Through this process, subglacial sediments and rocks may experience lowering of their water content, and their permittivity, but also an increase in the electrical conductivity of the remaining fluids. These competing processes can maintain unexpectedly high bed reflectivity within zones of basal freezing and lead to misinterpreting them as zones of basal melting.

Of course, it would be best to be able to use radar observations to constrain both the permittivity and the electrical conductivity of subglacial materials. One piece of observational evidence, the phase shift of the reflected wave, can be used to independently check if the electrical conductivity of sub-ice materials plays a role in controlling basal reflectivity. Figure 2A illustrates that as the electrical conductivity becomes either very large or very small, the phase shift angle is small in either case, thus limiting the ability to use the phase angle to determine if strong radar bed reflectivity is due to high permittivity or conductivity contrasts. Another potentially helpful approach is to take advantage of the fact that the low-loss reflection coefficient is frequency independent (Eq. 11) while the full version and the high-loss version retain frequency dependence (Eqs. 8 and 12). Within the typical range of linear radar frequencies used in glaciology (1-400MHz), this frequency sensitivity of the reflection coefficient is the highest at low frequencies (1-10 MHz) and at relatively low conductivities (0.001-0.1 S m$^{-1}$) (Figure 2B). As the conductivity of subglacial materials approaches that of highly conductive clay-rich sediments and seawater (>0.1 S m$^{-1}$), the amplitude reflection coefficient becomes increasingly less sensitive to frequency. Dual- and multi-frequency radar systems may, thus provide, a useful constraint on the presence or absence of conductive materials beneath ice (e.g., Rodriguez-Morales et al., 2013). It may be possible to take advantage of the fact that ice-penetrating radars are not single-frequency radars

but emit waves over some bandwidth around the center frequency (e.g., 100 MHz). Hence, the

frequency-dependence of bed reflection may be revealed by comparing the power-frequency content of

this reflection to the power-frequency distribution for the emitted wave or a strong englacial reflector.

Incorporation of electrical conductivity into interpretations of bed reflectivity will lead to somewhat

more complicated radioglaciological analyses as compared to the simplicity of the low-loss assumption

(e.g., Eq. 8 vs. Eq. 11). However, it has the potential to unlock underexplored avenues of

radioglaciological research, by enabling mapping of sub-ice geology (e.g., clay content) and fluid

salinity in sub-ice water reservoirs on Earth and other planetary bodies with ice cover (e.g., Mars and

Europa). This is difficult to accomplish using the traditional low-loss assumption (Eq. 8) given that the

electrical conductivity of water changes by orders of magnitude with changing salinity, but its

permittivity is only weakly dependent on solute content (e.g., Midi et al., 2014).  The approach

presented here offers practical tools that can be used in such investigations without the need to employ

complex analysis (e.g., Peters et al., 2005). Once electrical conductivity is considered, the treatment of

radar wave reflection becomes explicitly dependent on frequency (Eqs. 8 and 12). However, even the

relative permittivity of water, and by extension of water-bearing sediments and rocks, is frequency

dependent (e.g., Buchner et al., 1999; Arcone et al., 2008; Midi et al., 2014).

## 6 Conclusions

The assumption that radar reflection is generated at an interface between two lossless dielectric materials is certainly appealing, because it simplifies the problem to a contrast solely in permittivity (Eqs. 11) and

eliminates the dependence of reflectivity on radar frequency and electrical conductivity. However, our examination of the criterion for the lossless conditions, $\sigma >> \varepsilon\omega$ ($\psi << 1$), indicates that it is unrealistic for a wide range of common geologic materials for the range of linear radar frequencies (1-400 MHz) used in glaciology, planetary sciences, and GPR investigations. This is particularly the case for the low frequency radars (e.g., 2-5 MHz center frequency) used in glaciology and planetary science, for which

even materials with conductivity as low as ca. 0.0001-0.001 S m$^{-1}$ (1,000-10,000 $\Omega$m) are too high for the lossless criterion to be applicable (Fig. 2). But even at the high end of frequencies (ca. 100 MHz), a number of geologic materials can have high enough conductivity, 0.01-1 S m$^{-1}$ (1-100 $\Omega$m) for it to matter in radar reflectivity. In the absence of *a priori* constraints on the electrical conductivity of target materials, interpretations of radar interface reflectivity should be made based on the full form of the reflection

coefficient, which retains the dependence on conductivity and frequency, in addition to permittivity (Eq. 8). Since Eq. (8) contains at least two unknown material properties, the permittivity and the conductivity of the target material (e.g., subglacial material), it is possible to gain additional constraints using either the phase shift of the reflected wave (Eq. 10) or the frequency dependence of the reflection coefficient (Eqs. 8, 12). In some cases, for instance when ice is in contact with a body of water, sub-ice permittivity

is known and the basal radar reflectivity can be used to directly constrain the sub-ice electrical

conductivity, $\sigma_2$. This may allow estimating the salinity of subglacial lakes on Earth and sub-ice oceans on icy planetary bodies.

**7 Team list**

Slawek M. Tulaczyk and Neil T. Foley at the Department of Earth and Planetary Sciences, University of California, Santa Cruz.

**8 Author contributions**

Slawek M. Tulaczyk designed this research, performed analyses, and wrote the manuscript. Neil T. Foley co-designed this research and contributed to manuscript writing and editing.

**9 Competing interests**

The authors declare that they have no conflict of interest.

## 10 Acknowledgments

This material is based upon work supported by the National Science Foundation under Grant No. 1644187. The content of this paper is the sole responsibility of the authors. The authors are thankful to two anonymous reviewers for helpful comments that greatly improved the quality of this paper.

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

*Figure 1.* **(A)** Schematic diagram showing the incident radar wave, $E_o$ and solid arrow, the reflected

wave, $E_r$ and dashed line, as well as the transmitted wave, $E_t$ and the dotted arrow. The horizontal thick

line represents the reflective interface between materials 1 and 2, each characterized by three material properties: magnetic permeability, permittivity, and conductivity. The two grey horizontal dashed-dotted lines illustrate the two length scales relevant to wave reflection, the skin depth, $\delta$, and the half wavelength, $\lambda/2$. This figure is adapted from Stratton (1941, figure 96). **(B)** Plot of the phase constant, $\alpha$, and the attenuation constant, $\beta$, with the control parameter $\psi = \sigma/(\omega\varepsilon)$ on the horizontal axis and the pre-factor from Eq. (3ab), $\omega\mu^2\varepsilon^2/4$, on the vertical axis. The solid lines show the full version of the expressions 3ab while the dashed horizontal line represents the lossless approximation of the phase constant, $\alpha$ (Eq. 4a). The dashed diagonal line gives the high-loss version of the phase and attenuation constants, $\alpha$ and $\beta$, which are equal to each other (Eq. 5a). The two grey regions on the left- and the right-hand side of the figure shows, the low loss and high loss conditions, respectively, in which the lossless and the high-loss solutions represent reasonable approximations of the full solution. **(C)** Limits of lossless and high-loss conditions for three different linear radar frequencies, 1 MHz, 10 MHz, 100 MHz plotted in the conductivity-permittivity space. **(D)** The full version of the amplitude reflection coefficient, $r$ (Eq. 8), plotted for the case of 100 MHz linear frequency as a function of electrical conductivity, $\sigma_2$, and relative permittivity of the sub-ice material, $\varepsilon_r = \varepsilon_2/\varepsilon_o$. The relative permittivity is plotted at the increment of 5 between its assumed minimum value of 5 and the maximum value of 85. For ice, we use relative permittivity of 3.2 and the electrical conductivity of $10^{-5}$ S m$^{-1}$ (Stillman et al., 2013). The right-hand-side axis gives the power reflection coefficient, $R$ (Eq. 9), in decibels.

*Figure 2.* **(A)** An equivalent plot to Figure 1D but here the tangent of the phase shift angle (Eq. 10) plotted for the case of 100 MHz linear frequency as a function of electrical conductivity and relative

permittivity of the sub-ice material. The equivalent phase shift angles are given on the right axis. The material properties of ice are as assumed in Figure 1D.

**(B)** A plot demonstrating the frequency dependence of the high-loss version of the amplitude reflection coefficient, $r$ (Eq. 12), for different values of electrical conductivity of the sub-ice material. The material properties of ice are as assumed in Figure 1D. The right-hand-side axis gives the power reflection coefficient, $R$ (Eq. 9), in decibels.




**Figure 1**

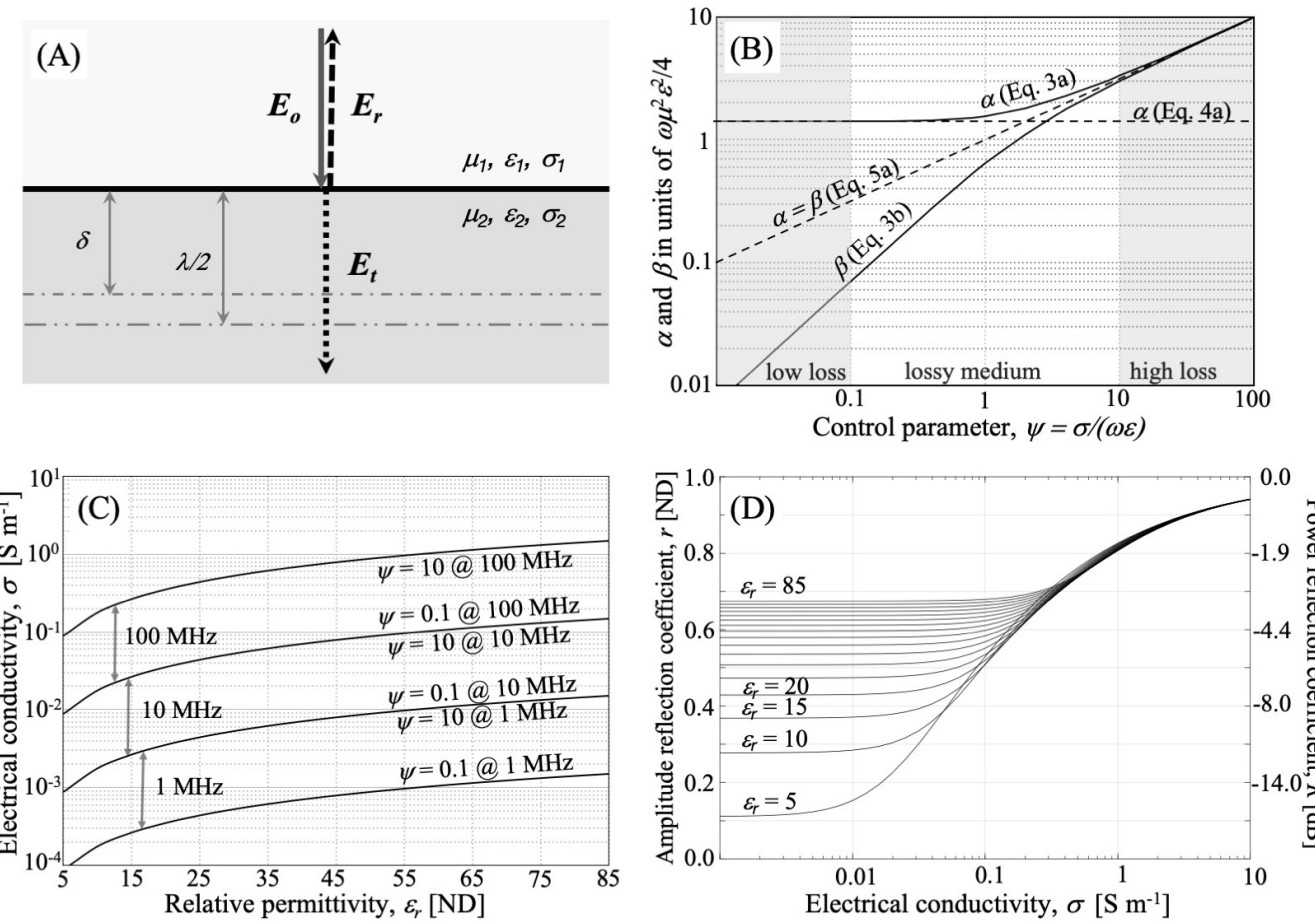

**Figure 2**

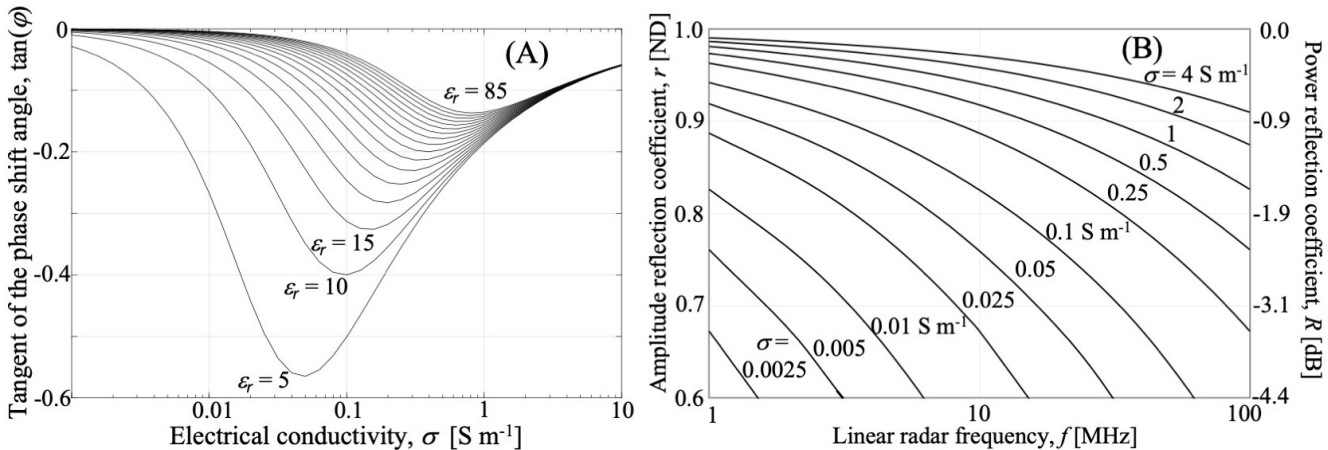