# Peer review of "The role of electrical conductivity in radar wave reflection from glacier beds"

_The Cryosphere, 2020_

## Referee Comment (RC1) · Anonymous Referee #1 · 24 Feb 2020

please see attached zip file

Please also note the supplement to this comment:
https://www.the-cryosphere-discuss.net/tc-2020-9/tc-2020-9-RC1-supplement.zip

---

## Referee Comment (RC2) · Anonymous Referee #1 · 25 Feb 2020

Dear authors,

Thank you for further clarifying some important points. I apologize that my review read more negatively than intended, and I think part of the confusion is due to how reviews are classified in TC.

I rejected and graded the article in its 'current form' as a full-length TC research article (as we have to do as reviewers) as I believed that the original content could be distilled to a few pages and $\sim$ 2 figures with subpanels. TC did not give the option to 'recommend resubmission as a brief communication' and, had this option been available, I would have chosen to do this. I thought this recommended action was evident in my review as I stated: 'Ultimately, I think the study would be better packaged as a brief communication/letter or needs to be significantly extended with more original material

to be a full-length TC article... I think the results are useful to the subfield and it would be good to see a substantially revised version of the study published in the future'.

From the statement above it should be also evident that I have no intention to block the underlying science or act unscientifically. I gave feedback with the hope of improving the final publication to better focus on points of novelty.

From your comments, I now appreciate that you consider there to be pedagogical value to presenting the complex wavenumber form of the reflection equation. As I was focused on demonstrating numerical equivalence (for assumed permittivity and conductivity for the different forms) I arguably overlooked this point. To my knowledge the complex wavenumber form has not been presented in the glaciology literature hence could justify a longer article to make the formulation accessible.

In which case my major comments could be revised/incorporated in a full-length article as follows:

1. Equivalence of two forms. It would be highly desirable, and consistent with the pedagogical flavor of the MS, to cross-reference between complex wavenumber and complex permittivity forms of the Fresnel equation (the latter form is what I believe most of the field are more familiar with which is why I recommend this as the starting point). This should also include a cross-reference to the loss tangent as it is already used implicitly as the control parameter on the x-axis of Fig. 1 to establish loss limits. (Note: for a lossy medium tanâĄą(delta)=$\varepsilon$"/$\varepsilon$' =$\sigma/(\varepsilon'\omega)$ which is what I used to demonstrate numerical equivalence of the 2 equations in my attached script). There are some radar analysis applications when the loss tangent arises as the natural variable to use to infer material properties; specifically when assessing losses through a layer of unknown permittivity.

2. Volume of original work. If we assume that communicating the complex wavenumber form to glaciologists has been justified as a contribution, then my point 2 could become redundant as sections 2 and 4 are now necessary. I had a read through to check I have

not overlooked anything, and I apologize that I potentially overlooked equation 12 as a new analytic form that isn't in the literature. If this result is new then it should be heavily emphasized, as the preceding formulae in Sect 2 and 4 are textbook results. Related to this point - graphically comparing equation 12 with the full/general formula would be useful (particularly in the context of Fig 4).

Re: So, I challenge RC1 to provide supporting evidence of her/his contention that our analysis is basically old news that is not worth publishing because the radioglaciology community already knows about it

If we are using citations as a metric then Peters et al. 2005 have 175 on google scholar and quotes the full complex form of the Fresnel equations (admittedly an extra step is required to convert loss tangents into conductivities). I took this paper as the reference point for what 'common knowledge' in our field is, as I know many use it as a mini textbook. To my eyes, the form in Peters also represents the logical starting point from a pedagogical perspective (though I appreciate this is subjective).

I did not suggest that the reflectivity parameter evaluation in Sect 5 was old news or not worth publishing. Instead, I gave evidence that it could be done using the permittivity formula in Peters et al. 2005 as a definition (hence significantly shortening the article to focus on the new content). I also agreed that the results in Fig. 4 were a valuable contribution: 'In particular; the result that conductive clay can be more reflective than freshwater for certain (lower) frequency ranges and conductivities stands out'.

I think my reference to Berry 1975 was used out of context by the authors in their response on this point (originality/common knowledge within subfield). I used it to make a subsidiary point about scattering loss that can occur from a 2-phase dielectric mixture (cross-referencing Peters 2005). This could (hypothetically at least) act to reduce reflection strength from saline till relative to freshwater.

Finally, Oswald 2008, 2018 and Jordan 2018 are all considering airborne systems (150 -195 MHz) which is why I thought it would be useful to briefly discuss the impact of

the study (if any) on higher frequency systems (and potentially the typical scientific questions addressed in airborne data analysis).

I hope these comments are useful and improve clarity.

---

## Referee Comment (RC3) · Anonymous Referee #1 · 25 Feb 2020

Dear authors,

Thanks again for your response. Again - this is helpful clarity that further improves my appreciation of the study and the approach you have taken when writing the MS.

Just to confirm: as you gave a justification for providing the background context in sections 2 and 4, I think you can continue to submit as a full-length article if you think it is best for your study. (I will officially notify the editor that my initial verdict was a misplaced way of recommending a brief communication in the TC system). Part of my rationale for recommending a brief communication was that I think it better serves for 'general commentary' of the novel aspects to your work and how it applies to literature in our field.

Re: airborne radar studies and neglect of conductivity

I do agree that the papers you have selected focus on interpreting basal reflectivity in terms of the real permittivity (despite sometimes quoting the full formula from Peters et al. 2005). My best explanation for why Oswald 2008, 2018 and Jordan 2018 do this, is that they are focused on ice-sheet-scale mapping of basal water/thaw (which in both papers required a set of simplifying approximations to move forward). What I still think needs to be developed in more detail is whether your new results will significantly alter how similar airborne/water mapping studies could be influenced in the future given the relative role of other terms in the radar power relation, and the slightly higher frequency of airborne systems.

Hopefully when the dust settles this will be an example where TC open discussions have been useful. I wish you all the best with the resubmission.

---

## Short Comment (SC1) · 25 Feb 2020

I thank RC1 for taking the time and reading our manuscript and for providing comments on it. In particular, I value the detailed feedback on other past publications, which incorporated the impact of electrical resistivity on radar wave reflection from glacier beds (e.g., Berry, 1975).

However, I am surprised, and deeply disappointed, by the conclusion of this reviewer that our manuscript is not suitable for publication in The Cryosphere. The Cryosphere, and similar journals like J. Glac., published recently a number of radioglaciology papers, which make the assumption that glacier bed reflectivity depends solely on relative permittivity, and hence water content, and make no mention of the fact that electrical

resistivity of subglacial clays and fluids may also play a role (e.g., Jordan et al., 2018; Oswald et al., 2018; Passalacqua et al., 2017). This assumption is most often not even stated in these papers, it is just taken as factual truth that bed reflectivity is simply a function of subglacial water content, through its impact on relative permittivity. Yet, RC1 is telling us that our manuscript is not worth publishing because it is basically old news, since the influence of electrical resistivity on glacier bed reflectivity has been considered by Berry (1975).

Well, if our message about the need to consider electrical resistivity in interpretations of radar bed reflectivity is old news, it is old news that few in the radioglaciology community seem to be hearing. This problem is illustrated by the fact that the paper of Berry et al. (1975) has been cited only 43 times since 1975 (less than once per year) and seemingly dropped off the 'radar' of the community since it only got two citations since 2016 (based on Google Scholar). In contrast, the paper of Oswald and Gogineni (2008), which is based on the assumption that bed reflectivity has a one-to-one relation to bed wetness (their figure 3) and makes no mention of either electrical resistivity or conductivity of subglacial materials, has been cited 93 times in the 12 years since its publication and 42 times since 2016 (again, based on Google Scholar). It seems to me that the radioglaciology community is in a dire need of a reminder that electrical conductivity, in addition to relative permittivity, influences radar reflectivity of glacier beds.

We make no claims in our manuscript that we derive a new relationship for radar reflectivity of glacier beds. Clearly, we go back to a textbook from 1941 to show that such an expression can be easily obtained from Stratton's theory of radar wave reflection. However, what we do offer in our manuscript is an analytical expression for radar reflectivity as a function of permittivity and conductivity that is not wrapped up in complex numbers and abstract concepts such as the loss tangent. Clearly, some in the radioglaciology community (e.g., Berry, 1975 and Peters et al., 2005) have considered the impact of electrical conductivity on bed reflectivity, but the complex math associated

with these treatments has been a barrier to a general uptake of this approach in the community. Instead, even the recent relevant publications make statements like: "... the reflection coefficient, which varies with the relative permittivities above and below the interface ..." (Oswald et al., 2018), and "The relative (real) part of the permittivity is the primary control on [R]." (Jordan et al., 2018), where [R] in the latter is the reflection coefficient. I can easily provide dozens of examples of radioglaciology papers in which their authors make zero mention of the fact that electrical conductivity/resistivity may have an impact on radar reflectivity, in addition to the relative permittivity. But I know of only a handful of papers that do consider electrical conductivity/resistivity when radar bed reflectivity is interpreted (e.g., Peters et al., 2005). So, I challenge RC1 to provide supporting evidence of her/his contention that our analysis is basically old news that is not worth publishing because the radioglaciology community already knows about it.

Our manuscript provides an approachable formulation for the radar reflectivity that does not require users to deal with complex and imaginary numbers. Anybody with an Excel spreadsheet or rudimentary coding skills can use our work to quickly get at the dependence of glacier bed reflectivity on permittivity and electrical conductivity/resistivity. Effective communication of quantitative concepts is as important as the concepts themselves. It is not a coincidence that radioglaciologists prefer to interpret bed reflectivity in terms of relative permittivity and wetness, it is simply because they are avoiding having to deal with the complex math concepts involved in the past formulations that included electrical conductivity. In this manuscript we remove this barrier to considering electrical resistivity in interpretations of radar bed reflectivity.

We also provide graphical illustration of the dependence of radar reflectivity on electrical conductivity (our figure 4). I am pretty familiar with dozens of papers in the radioglaciology literature and I do not recall that anybody before has produced a figure even similar to ours. Contrast our figure 4 with the figure 3 in Oswald and Gogineni (2008) where bed reflectivity is simply plotted as a single line expressing the assumed one-to-one relationship between bed reflectivity and relative permittivity (i.e., bed wetness). In

our manuscript we carefully consider the criteria under which one is, or is not, justified to ignore electrical conductivity of subglacial materials (the low-loss and high-loss conditions). We also point out that many geologic materials, particularly clay-bearing rocks and sediments, have high enough electrical conductivity for their conductivity to matter in determining the radar reflection coefficient. Our manuscript points out that high conductivity subglacial materials (e.g., clays, brines, conductive minerals), can produced some of the brightest radar reflections (>90% amplitude reflection), which are brighter than the ca. 67% amplitude reflection from ice-water interface in the high-loss assumption. I have seen many examples, in literature and during conference presentations, of radioglaciologists claiming that ice on top of water bodies (e.g., subglacial lakes) should produce the strongest radar reflection. We point out that a patch of wet clay-bearing sediments can be brighter than a subglacial lake with fresh meltwater.

We also point out in our manuscript that the dependence of radar reflectivity on electrical conductivity can actually be useful in practical applications. For instance, under the common, low-loss assumption there should be no difference in radar reflection from ice overlying a freshwater lake versus a high salinity, briny lake. However, when electrical conductivity is taken into account, the radar reflectivity of the latter can be as much as 30% higher than that of the former. I have recently had a discussion with a pair of experienced Europa researchers and they were surprised to find out that a future radar mission to Europa may be able to provide constraints on the salinity of Europa's ocean, due to the sensitivity of radar reflectivity to electrical conductivity. This is because they, just as most of the terrestrial radioglaciology community, were under the assumption that only contrasts in relative permittivity matter in determining the strength of radar reflection from an ice-water interface.

Given all of the above, I find it surprising that the reviewer is recommending rejection of our manuscript. What we write about is not 'old news' that is well internalized by the radioglaciology community. And we clearly make contributions that have not been made by others before. It is simply unscientific to try to silence us just because we

are trying to point out that an assumption that is commonly made in radioglaciology papers may not be justified in general. Whether the publication of our manuscript will be blocked or not, this will not change the fact that relative permittivity is, in general, not the sole control on radar reflectivity of glacier beds. It is just a convenient, but not universally applicable, simplifying assumption that has been made so many times that many in the community think that it is based on a universal law of physics.
* * *

---

## Short Comment (SC2) · 25 Feb 2020

Thank you for further clarifying your review of our manuscript. If at all possible, given your review and the review that will come from the second reviewer, we will re-format our manuscript into the shorter TC format.

One issue I want to follow up on is the issue of Peters et al. (2005) being widely cited in the radioglaciology literature. It is certainly true that this paper is widely cited and that it includes a complex reflection coefficient with electrical conductivity taken into account through the loss tangent term (their table 1 and equations 6 and 7). However, this does not mean that all papers citing Peters et al. (2005) take electrical conductivity into account when calculating radar reflectivity. Just the opposite, only a few papers

citing Peters et al. (2005) seem to do that (e.g., MacGregor et al., 2011; Christianson et al., 2016). I went back to the five recent papers on radar reflectivity of glacier beds that I read when writing this manuscript (Jordan et al., 2017, 2018; Chu et al., 2018; Oswald and Gogineni, 2008; Oswald et al. 2018) and none of them makes use of the loss tangent values for subglacial materials from table 1 of Peters et al. (2005). Jordan et al. (2018) uses the table 1 from Peters et al. (2005) but only copies their values for relative permittivity and does not use the loss tangent values given in the table. They also use the real version of the reflection coefficient (the one we derive using the low-loss assumption in our manuscript), not the complex one given by equations 6 and 7 in Peters et al. (2005).

Somehow, the radioglaciology community uses Peters et al. (2005) a lot but seems to overlook the fact that this paper clearly points out the need to consider both, relative permittivity and electrical conductivity (or at least its representation in the form of the loss tangent) in calculations of radar reflectivity. This 'blind spot' is particularly perplexing since the same community is used to dealing with the fact that electrical conductivity causes radar wave attenuation in ice. For instance, Oswald et al. (2018) use the loss tangent concept to treat attenuation in ice but their expression for radar reflectivity (equation 7) is only based on relative permittivity of subglacial materials, although such materials can be many orders of magnitude more conductive than glacier ice. Again, I believe that this is because few want to deal with complex numbers in Peters et al. (2005) formulation of the reflection coefficient. We offer an alternative approach that avoids this problem. With our equations any radioglaciologist can calculate reflection coefficients using just real values of electrical conductivity and permittivity.

All scientific papers have pedagogical dimension. We are trying to teach each other something about how Nature works. One of the reviewers of my early paper on till deformation (Tulaczyk et al., 2000) criticized that manuscript as being too pedagogical because we were using basic concepts from soil mechanics to help explain observed features of subglacial till deformation. Yet, this paper has been cited nearly 400 times,

more than any Science or Nature paper that I have been part of. Apparently, the community appreciates if you teach them something useful.

---

## Short Comment (SC3) · 26 Feb 2020

I am certainly enjoying this opportunity to have a back-and-forth discussion with a reviewer. Thank you for your continuing feedback.

With regards to your comment on ice-sheet-wide studies of bed wetness making the simplifying assumption that electrical conductivity does not influence bed reflectivity, I am not sure why the scale of a study should matter here (other than just for pure convenience). Actually, the broader the geographic region, the more variability in electrical conductivity of subglacial materials one would expect because a larger region is more likely to have significant variations in geology (e.g., clay content of subglacial materials) and in the conductivity of subglacial water (due to changing solute content) than a

smaller region. In reality, it means that these authors are interpreting spatial variations in bed reflectivity that are partly caused by spatial variations in subglacial electrical conductivity as being caused by spatial variations in subglacial water content. Areas where ice is underlain by clay-bearing till, or subglacial shale, or even subglacial schist (or any rock/sediment saturated with brine), are being misinterpreted as areas containing a lot of subglacial water.

Furthermore, one of the more common uses of bed reflectivity is to distinguish areas of frozen and melted conditions. Yet, basal freezing can lead to cryoconcentration of solutes in the remaining subglacial waters (e.g., Badgeley et al., 2017, J.Glac.) Cryoconcentrated solutions have higher electrical conductivity. This means that the relative abundance of subglacial water can be shrinking over time but the radar reflectivity of the bed may not be changing much because the smaller amount of remaining water is more conductive, and hence more reflective than one would infer just from the relative permittivity of the material. In addition, the electrical conductivity of clay-rich sediments and rocks is going to change very little with changes in their water content. So, over such materials bed reflectivity will be changing very little with changes in their water content. Here I should note that both Peters et al. (2005) and Christianson et al. (2016) in their tables giving the loss tangent / electrical conductivity are approximating subglacial till as a mixture of sand and water. This is not a good approximation since glacial tills are famous for being mixtures of fine-grained matrix (clay and silt) with pebbles/cobbles/boulders (hence, the traditional name for glacial tills is 'boulder clay'). This means that Peters et al. and Christianson et al. have underestimated how electrically conductive glacial tills can be because finer-grained sediments are more conductive due to their larger specific surface area.

I just think that the radioglaciology community, by simplifying the reflection coefficient to be just a function of subglacial relative permittivity (and hence bed water content), is missing an opportunity to study subglacial conditions in a more realistic way. By recognizing that electrical conductivity plays a role in controlling the reflection coefficient,

one will open a door to also mapping out geologic conditions beneath ice, something that is poorly constrained but important to ice sheet evolution.

---

## Referee Comment (RC4) · Anonymous Referee #2 · 6 Mar 2020

Tulaczyk and Foley (2020) derive the reflection coefficient for the low loss region, the high loss region, and provide the general reflection coefficient that describes the regions of conductivity that are in between. They highlight the impact that conductivity has on the reflection coefficient, and note that for highly conductive materials with low permittivity values, one could obtain reflection coefficients that are even greater than for a pure water subglacial lake. This is a key point for the glaciological community to understand, as one often attributes a brightly reflecting subsurface as subglacial water; however, Tulaczyk and Foley raise the concern that instead of attributing a strong bed echo directly to subglacial water, we should look for additional constraints such as the phase of the returned signal before making such an assertion, because "subglacial sediments can be conductive enough to produce radar reflectivity that is the same, or

higher, than reflectivity from an ice-lake interface", even if they have a lower relative permittivity.

In addition to reminding the community of the significance of conductivity in radar reflection, the article was clear to read. The reminder of the resistive nature of ice and the role that conductivity plays in radar signal attenuation and reflectivity is appreciated since it is an important aspect of the material property that often gets overlooked. However, I do have some concerns about the novelty aspect of the manuscript.

Major Comments

For example, even though it has been adapted to include the conductivity, one can find very similar versions of Figure 3 online with a quick google search of "plane wave at the media boundary" (https://www.brainkart.com/media/extra/VZti9GN.jpg). Furthermore, some of the main equations of the text - Equations 7(ab), Equations 8-10 – one could get straight from Stratton (1941) and Equation 11 could come straight from Peters 2005. Finally, Table 1 and Equations 6 and 7 of Peters (2005) already include the conductivity (in the loss tangent column) and show how it produces a complex reflection coefficient (by combing the reflection strength and phase columns). It's hard to see how the results or analyses differ from these published results?

Minor Comments

It would be useful to see Figure 2 and Figure 6 go out to center frequencies greater than 100 MHz. For example, MCoRDS operates from 140 to 230 MHz and ApRES radar operates between 200 to 400MHz. It would be interesting to see the limits of lossless and high-loss conditions for linear radar frequencies up to around 400 MHz.

The authors may want to consider combining Figures 3, 4, 5, and 6 into a 4-panel figure as they seem to go together and Figure 3 by itself is slightly underwhelming. At a minimum, Figures 3-5 are related and could be combined into a paneled figure.

Line-by-Line Specific Comments

[Figure]

Line 45 the authors note that "inferences about sub-ice conditions, e.g., the presence or absence of subglacial water, are drawn from the lateral variations in radar bed reflectivity." I would also add that these inferences of subglacial water could come from temporal variations in reflectivity; for example, a stationary ApRES system that is deployed for a year could monitor the reflectivity changes at a single location.

Line 53 insert "the" to read as "In general, the mathematical"

Line 229 insert "of" to read as "system of two equations"

Line 230 insert "the" to read as "illustrate the limitations"

Line 230, Line 264, and Line 302, "regimens" should be "regimes"

Line 236 Equation 11. I would again note to the reader that we are assuming that both of the media have the magnetic permeability of free space

Line 246 Equation 12, in both the numerator and the denominator, the term after $(\alpha\_1-\alpha\_2)$ˆ2 should be $\beta\_2$ˆ2, not $\alpha\_2$ˆ2

Line 335 at the very end, remove the hyphen after the comma but before "interpretations"

Line 340 "reflected way" should be "reflected wave"

Line 343 modify "will allow estimating" to "will allow one to estimate"

In Figure 4 caption, "10-5" should be superscript, 10ˆ-5

While the authors note they "plotted for the case of 100 MHz linear frequency" in the main text for Figure 4, I would also note this point in the captions of Figure 4 and Figure 5.

In Figure 6, I would denote the electrical conductivity symbol for each line (i.e. $\sigma$=4 S m-1) similar to what is done for Figures 4 and 5 showing their relative permittivity symbols for each line (i.e. Ït_r=85, etc.).

---

## Short Comment (SC4) · 7 Mar 2020

Thank you for your review of our manuscript and for your numerous helpful specific suggestions. I do have a few clarifying points and a few questions:

(1) FIGURE 3 - I am somewhat perplexed by your criticism that our Figure 3 looks like a figure that can be found online. This figure is in the manuscript just to graphically illustrate the physical setup that we are discussing (i.e., two media with different EM properties separated by a flat interface with an incident EM wave reflecting from, and transmitting through, this interface). Anybody who talks about EM wave reflection from a flat interface separating two homogeneous, isotropic media will draft a similar figure. There is not really much room, or need, here for creativity. Our figure is just a slightly

modified version of a figure in Stratton. Can you be more specific as to what would constitute a more acceptable figure? If you prefer that we modify the figure that you found online, instead of using the one we have in the manuscript right now, we can sure do that.

(2) EQUATIONS 7a THROUGH 11 - We are not hiding the fact that some of the equations in the manuscript come straight out of Stratton's book. In fact, wherever we did take an equation straight from the book, we provided a reference to the page and equation number. Contrary to your assertion, out of the equations called out in your review (Equations 7ab, 8, 9, 10, 11), only some come directly from Stratton's book. 7a does but 7b is obtained by us from 7a by setting the magnetic permeability to be the same for both media and by writing out the propagation constants in terms of the attenuation factor and the phase constant. Maybe somewhere in Stratton's book there is an equation that is exactly the same but we got 7b from 7a. If you can provide a page number and an equation number where you found 7b in Stratton's book, I'll be happy to cite it. Similarly, we did not copy equation 8 from Stratton's book. We did take the absolute value of the complex reflection coefficient to obtain this equation. As with equation 7b, if you can provide a page number and an equation number where you found 8 in Stratton's book, I'll be happy to cite it. I could not find this form of the equation in the book but may have missed it. We quote the book pages and equation numbers were we found 9 and 10. The point about deriving equation 11 is to demonstrate that one can obtain the commonly used version of the radar reflection coefficient from equation 8. If equation 8 would not simplify to 11 under the low-loss assumption, this would cast doubt on the validity of equation 8. We show equation 11 as a reassurance that the more complete version of the radar reflection coefficient shown in equation 8 does simplify to the commonly used form (e.g., Peters et al., 2005 and many others) when the reflection is assumed to take place from a boundary between two perfect dielectrics. So, the very point of deriving equation 11 is to show that 11 has the widely used form, not to claim that we have re-invented the wheel. I will change the discussion of equation 11 to make this clearer.

(3) THIS HAS BEEN DONE BY PETERS ET AL. 2005 - Our manuscript provides an approachable formulation for the radar reflectivity that does not require users to deal with complex and imaginary numbers. Anybody with an Excel spreadsheet or rudimentary coding skills can use our work to quickly get at the dependence of glacier bed reflectivity on permittivity and electrical conductivity/resistivity. Effective communication of quantitative concepts is as important as the concepts themselves. It is not a coincidence that radioglaciologists prefer to interpret bed reflectivity in terms of relative permittivity and wetness, it is simply because they are avoiding having to deal with the complex math concepts involved in the past formulations that included electrical conductivity through the obscure concept of loss tangent. By now there are many decades of published geophysical measurements of subsurface electrical conductivity/resistivity using a variety of techniques. These values can be used by radioglaciologists. Loss tangent is a much more abstract, and less frequently used, concept in geophysics. In this manuscript we remove barriers to considering electrical resistivity in interpretations of radar bed reflectivity. We also provide graphical illustration of the dependence of radar reflectivity on electrical conductivity (our figure 4). I am pretty familiar with dozens of papers in the radioglaciology literature and I do not recall that anybody before has produced a figure even similar to ours. Contrast our figure 4 with the figure 3 in Oswald and Gogineni (2008) where bed reflectivity is simply plotted as a single line expressing the assumed one-to-one relationship between bed reflectivity and relative permittivity (i.e., bed wetness). In our manuscript we carefully consider the criteria under which one is, or is not, justified to ignore electrical conductivity of subglacial materials (the low-loss and high-loss conditions). We also point out that many geologic materials, particularly clay-bearing rocks and sediments, have high enough electrical conductivity for their conductivity to matter in determining the radar reflection coefficient. Our manuscript points out that high conductivity subglacial materials (e.g., clays, brines, conductive minerals), can produce some of the brightest radar reflections (>90% amplitude reflection), which are brighter than the maximum reflection of ca. 67% amplitude reflection from ice-water interface in the high-loss assumption.

I have seen many examples, in literature and during conference presentations, of radioglaciologists claiming that ice on top of water bodies (e.g., subglacial lakes) should produce the strongest radar reflection. We point out that a patch of wet clay-bearing sediments can be brighter than a subglacial lake with fresh meltwater. We also point out in our manuscript that the dependence of radar reflectivity on electrical conductivity can actually be useful in practical applications. For instance, under the common, low-loss assumption there should be no difference in radar reflection from ice overlying a freshwater lake versus a high salinity, briny lake. However, when electrical conductivity is taken into account, the radar reflectivity of the latter can be as much as 30% higher than that of the former. I have recently had a discussion with a couple of experienced Europa researchers and they were surprised to find out that a future radar mission to Europa may be able to provide constraints on the salinity of Europa's ocean, due to the sensitivity of radar reflectivity to electrical conductivity. This is because they, just as most of the terrestrial radioglaciology community, were under the assumption that only contrasts in relative permittivity matter in determining the strength of radar reflection from an ice-water interface. Given all of the above, I contend that what we write about is a much more complete treatment of the role of electrical conductivity in bed reflection strength than what is presented in Peters et al. (2005). We clearly make contributions that have not been made by others before. Relative permittivity is, in general, not the sole control on radar reflectivity of glacier beds. Anybody using a version of our Equation 11 when interpreting radar bed reflection strengths is simply making a convenient, but not universally applicable, simplifying assumption that has been made so many times that many in the community think that it is based on a universal law of physics.

We will follow your specific suggestions for improving the manuscript (e.g., considering linear frequencies higher than 100 MHz.)

---

## Author Comment (AC1) · 30 May 2020

The handling editor, Dr. Nanna Karlsson, informed us that at this stage she does not need a line-by-line response to reviewers' comments because we had a back-and-forth discussion with them. The editor did, however, request that we outline the actions intended to incorporate the reviewers' suggestions in the revised version of this manuscript. We do so in the text provided below.

We have analyzed the helpful feedback provided by the two reviewers and propose the following major changes to the revised manuscript to improve it in a way consistent with reviewers' suggestions. Besides these major changes we will also make any minor improvements suggested by the reviewers (e.g., fixing typos, etc.)

[Figure]

(1) After considering the possibility of re-formatting our manuscript into the format of a TCD brief communication, we think that this format is too restrictive to allow us to improve the manuscript based on the feedback from the reviewers. Only up to 20 references are allowed by this format, and we already have more than 20 references cited in our manuscript. Besides, responding to reviewers' feedback will require additional references, not fewer references (see below). This manuscript will be most useful to the community if it is published as a regular paper, not a brief communication. Reviewer 2 commended the readability of our manuscript, so we would rather not cut its length drastically. Reviewer 1 also appears to have ultimately agreed that an extended version will be more useful than a brief communication. This manuscript needs to be readable because it deals with issues that are relatively abstract and mathematical. Moreover, we have to have sufficient space to make a compelling case that the conventional thinking about the meaning of bed reflectivity is too narrow. However, we will combine some of our figures into a single figure with multiple panels, to save space.

(2) We believe that the best way to address several reviewers' comments, is to include in the revised version of this manuscript a table of resistivity/conductivity values for glacial materials (glacial sediments, basal ice, etc.) This table will be an updated version of Table 1 published by Christianson et al. (2016). Their values for electrical conductivity / resistivity of glacial materials were largely based on Looyenga's dielectric mixing model (Looyenga, 1965). However, we will use values either observed in laboratory experiments (e.g., Arcone et al., 2008; Josh and Clennell, 2015) or in the field (e.g., Foley et al., 2015; Mikucki et al., 2015; Jorgenesen et al., 2012; Steuer et al., 2009). This table will communicate to the radioglaciology community observationally constrained ranges of electrical conductivity for glacial materials. These ranges will be considerably broader than those used in prior similar compilations (e.g., Table 1 in Christianson et al., 2016). This table will support our contention that radioglaciological interpretations of subglacial materials need to consider the impact of electrical resistivity of such materials on the radar reflection from ice beds. We propose to include this new table in section 3 (Low-loss assumption and its limitation). We will modify the text

of this section to discuss this new table.

(3) In response to reviewers' concerns that our manuscript is not relevant or needed, we want to give more pertinent examples from recent radioglaciological studies. These examples will review how the published studies would have benefited from considering the possibility that the electrical conductivity of subglacial materials influences the strength of the bed reflection coefficient. In particular, both of the examples we used for this purpose in our Discussion section, come from one specific part of Antarctica. In the revised version of the manuscript, we will draw on recent work in Greenland (e.g., Chu et al., 2018; Jordan et al., 2017, 2018; Oswald and Gogineni, 2008; Oswald et al., 2018). We will expressly point out how the conclusions of these papers may have changed if the authors would consider the impact of electrical conductivity on bed reflectivity. We will work integrating our observations into a table, or a figure, that will provide a more concise and more graphically compelling summary of this part of our manuscript. These revisions will significantly modify the Discussion section of the manuscript.

(4) Finally, we believe that the revised manuscript will benefit from expanding the Discussion section to include a paragraph explaining that consideration of electrical conductivity in interpretations of the bed reflection coefficient will be constructive. For instance, this approach will help map subglacial geology and may enable the determination of the salinity of subglacial lakes and oceans on icy planetary bodies. The latter is now only mentioned in passing in the conclusions, but it should be discussed more fully in the revised version.

---

## Author Response (AR1)

**UNIVERSITY OF CALIFORNIA, SANTA CRUZ**

[Figure]

Dr. Slawek Tulaczyk
Department of Earth and Planetary Sciences
A109 Earth and Marine Sciences Building
University of California
Santa Cruz, CA 95064, USA

(831) 459-3074 (fax)
stulaczy@ucsc.edu

07/27/2020

Dr. Nanna Karlsson
Editor, The Cryosphere

Dear Dr. Karlsson,

On behalf of myself and my co-author, Dr. Neil Foley, I submit a revised version of our manuscript (tc-2020-9). We are very grateful to you and to the two reviewers for the many insightful comments which guided our efforts aimed at improving this manuscript during the revision process.

In our past communications you indicated that you consider the responses to reviewers' comments that we posted previously to the TCD manuscript tracking system as sufficient point-by-point responses to these comments. Hence, I will focus in this letter on summarizing the manuscript changes that we made to allay the concerns that the two reviewers had about the original version of our manuscript.

One of the suggested changes that we did not implement was the idea that we should turn this manuscript into a TC Brief Communication. Perhaps the most restrictive part of this format is the fact that we could only cite 20 reference. Even our original manuscript exceeded this limit and you can see in the attached marked-up manuscript that we added more citations. Although I generally agree that, when possible, it is better to communicate science through brief publications. However, in this case I believe that such shortening of the manuscript would make it impossible for us to properly address the many insightful comments provided by the reviewers.

Here are the changes we did implement by manuscript section:

1. Introduction - In response to concerns expressed by the reviewers that our work is not different than past treatment of electrical conductivity as a control on bed reflectivity (e.g., Peters et al., 2005), we now state explicitly in the introduction that the primary difference is that the equations for radar reflectivity presented in our manuscript do not use complex numbers. We also comment that, in our opinion, our approach removes a significant barrier to wider consideration of electrical conductivity in radioglaciology research.

2. Section 2 (Background) - To improve communication, we introduce a control parameter ($\psi = \sigma/(\varepsilon\omega)$) which is used in Equations 3ab, in some figures, and in discussions. This parameter reflects the relative importance of electrical conductivity in equations used in this manuscript. We also provide a discussion of the physical meaning of this parameter and point out that its square root is proportional to a ratio of two length scales relevant to wave reflection, the half wavelength and the skin depth. Hence, the control parameter gauges the relative importance of displacement and conduction currents in the process of wave reflection from an interface separating media with non-zero electrical conductivity. We hope that the reviewers will find this addition to be informative and useful and that it will help allay their concern that our analysis is not adding much to what is already generally known.

3. As suggested by both reviewers, we have combined the first four figures from the original manuscript and created the new Figure 1 with four panels. The general schematic of the reflecting interface, the incident energy, the transmitted energy, and the reflected (now Figure 1A) was criticized by reviewer 2 as not being informative enough. We added to it the skin depth and the half wavelength and hope that it is now more acceptable to the reviewer.

4. We also combined the old Figure 5 and 6 into the new Figure 2 with two panels.

5. We added Table 1 at the end of Section 3 to provide a more comprehensive overview of relative permittivities and electrical conductivities for a range of basal and subglacial materials. The sources of these values are detailed in the extensive footnotes to this table and include past similar compilations, laboratory test results, and field measurements. We believe that this table provides a useful update to the similar compilation tables published in Peters et al. (2005) and Christianson et al. (2016). In Table 1 we also include the reflectivities corresponding to these material properties of glacial materials and use some of these reflectivities in subsequent discussions. Our intent here is to allay reviewers' concern that our manuscript is not adding enough to this discussion when compared to Peters et al. and Christianson et al. and MacGregor et al..

6. We made considerable additions to Section 5 (Discussion) with the aim of making a stronger case that radioglaciologists need to consider the impact of electrical conductivity on radar bed reflectivity. This is in response to the concern of both reviewers that our manuscript is not addressing any real need in the community. Hence, we expanded our arguments on the misinterpretations that may result from ignoring the impact of electrical conductivity on radar reflectivity in several radioglaciological applications: (i) mapping of ponded subglacial water, (ii) mapping of subglacial conditions near grounding lines, (iii) mapping of zones of basal freezing and basal melting. We added several citations to strengthen our case.

7. Towards the end of Section 5 (Discussion) we expanded our argument that there may be practical approaches to using radar data to constrain both permittivity and electrical conductivity of subglacial materials. In particular, we point that the power-frequency content of the bed reflection, when compared to the power-frequency content of the emitted pulses (or of englacial reflectors) may help constrain if the ice base reflects radar waves like an interface between two perfect dielectrics or not.

8.  We close Section 5 (Discussion) with a new paragraph, which re-iterates that inclusion of electrical conductivity in analysis of radar bed reflectivity opens new research possibilities (e.g., mapping of fine grained sediments or of areas with subglacial waters having high solute content). We also, again, point out here that our intention is to provide easy-to-use mathematical tools for including electrical conductivity alongside permittivity as a parameter in radioglaciological analysis. These additions, as well as those described above (point 7) are aimed at, again, convincing the reviewers that our manuscript has something to contribute to the radioglaciology community.

I am looking forward to reading reviews of our revised manuscript.

Please let me know if you have questions.

Yours sincerely,

Dr. Slawek Tulaczyk

Professor of Earth Sciences

Affiliate Faculty in the Environmental Studies and in Digital Arts and New Media

Fellow of the Geological Society of America

[revised manuscript text omitted]

Font: (Default) Times New Roman, 12 pt

| Page 25: [2] Formatted | Slawomir Tulaczyk | 7/28/20 9:10:00 PM |
| --- | --- | --- |

Font: (Default) Times New Roman

| Page 25: [3] Formatted | Slawomir Tulaczyk | 7/28/20 9:10:00 PM |
| --- | --- | --- |

Font: (Default) Times New Roman, 12 pt

| Page 25: [4] Formatted | Slawomir Tulaczyk | 7/28/20 9:10:00 PM |
| --- | --- | --- |

Font: (Default) Times New Roman

| Page 25: [5] Formatted | Slawomir Tulaczyk | 7/28/20 9:10:00 PM |
| --- | --- | --- |

Font: (Default) Times New Roman, 12 pt

| Page 25: [6] Formatted | Slawomir Tulaczyk | 7/28/20 9:10:00 PM |
| --- | --- | --- |

Font: (Default) Times New Roman

| Page 25: [7] Formatted | Slawomir Tulaczyk | 7/28/20 9:10:00 PM |
| --- | --- | --- |

Font: (Default) Times New Roman, 12 pt

| Page 25: [8] Formatted | Slawomir Tulaczyk | 7/28/20 9:10:00 PM |
| --- | --- | --- |

Font: (Default) Times New Roman, 12 pt, Not Italic

| Page 25: [9] Formatted | Slawomir Tulaczyk | 7/28/20 9:10:00 PM |
| --- | --- | --- |

Font: (Default) Times New Roman, 12 pt

| Page 25: [10] Formatted | Slawomir Tulaczyk | 7/28/20 9:10:00 PM |
| --- | --- | --- |

Font: (Default) Times New Roman

| Page 25: [11] Formatted | Slawomir Tulaczyk | 7/28/20 9:10:00 PM |
| --- | --- | --- |

Font: (Default) Times New Roman, 12 pt

| Page 25: [12] Formatted | Slawomir Tulaczyk | 7/28/20 9:41:00 PM |
| --- | --- | --- |

Font: (Default) +Body (Times New Roman)

| Page 25: [13] Formatted | Slawomir Tulaczyk | 7/28/20 9:41:00 PM |
| --- | --- | --- |

Font: (Default) +Body (Times New Roman), 12 pt

| Page 25: [14] Formatted | Slawomir Tulaczyk | 7/28/20 9:41:00 PM |
| --- | --- | --- |

Line spacing:  Double

| Page 25: [15] Formatted | Slawomir Tulaczyk | 7/28/20 9:41:00 PM |
| --- | --- | --- |

Font: (Default) +Body (Times New Roman), 12 pt

| Page 25: [16] Formatted | Slawomir Tulaczyk | 7/28/20 9:41:00 PM |
| --- | --- | --- |

Font: (Default) +Body (Times New Roman), 12 pt

| Page 25: [17] Formatted | Slawomir Tulaczyk | 7/28/20 9:41:00 PM |
| --- | --- | --- |

Font: (Default) +Body (Times New Roman), 12 pt

| Page 25: [18] Formatted | Slawomir Tulaczyk | 7/28/20 9:41:00 PM |
| --- | --- | --- |

Font: (Default) +Body (Times New Roman), 12 pt

| Page 25: [19] Formatted | Slawomir Tulaczyk | 7/28/20 9:43:00 PM |
| --- | --- | --- |

Font: (Default) +Body (Times New Roman), 12 pt, Not Italic

| Page 25: [21] Formatted | Slawomir Tulaczyk | 7/28/20 9:41:00 PM |
|---|---|---|

Font: (Default) +Body (Times New Roman), 12 pt

| Page 25: [22] Formatted | Slawomir Tulaczyk | 7/28/20 9:41:00 PM |
|---|---|---|

Font: (Default) +Body (Times New Roman)

| Page 25: [23] Formatted | Slawomir Tulaczyk | 7/28/20 8:52:00 PM |
|---|---|---|

Font: (Default) Times New Roman, 12 pt

| Page 25: [24] Formatted | Slawomir Tulaczyk | 7/28/20 8:32:00 PM |
|---|---|---|

Line spacing:  Double

| Page 25: [25] Formatted | Slawomir Tulaczyk | 7/28/20 8:52:00 PM |
|---|---|---|

Font: (Default) Times New Roman, 12 pt, Not Italic

| Page 25: [26] Formatted | Slawomir Tulaczyk | 7/28/20 8:52:00 PM |
|---|---|---|

Font: (Default) Times New Roman, 12 pt

| Page 25: [27] Formatted | Slawomir Tulaczyk | 7/28/20 8:52:00 PM |
|---|---|---|

Font: (Default) Times New Roman, 12 pt, Not Italic

| Page 25: [28] Formatted | Slawomir Tulaczyk | 7/28/20 8:52:00 PM |
|---|---|---|

Font: (Default) Times New Roman, 12 pt

| Page 25: [29] Formatted | Slawomir Tulaczyk | 7/28/20 9:08:00 PM |
|---|---|---|

Font: (Default) Times New Roman, 12 pt

| Page 25: [30] Formatted | Slawomir Tulaczyk | 7/28/20 9:08:00 PM |
|---|---|---|

Font: (Default) Times New Roman

| Page 25: [31] Formatted | Slawomir Tulaczyk | 7/28/20 9:08:00 PM |
|---|---|---|

Font: (Default) Times New Roman, 12 pt

| Page 25: [32] Formatted | Slawomir Tulaczyk | 7/28/20 9:08:00 PM |
|---|---|---|

Font: (Default) Times New Roman

| Page 25: [33] Formatted | Slawomir Tulaczyk | 7/28/20 9:08:00 PM |
|---|---|---|

Font: (Default) Times New Roman, 12 pt

| Page 25: [34] Formatted | Slawomir Tulaczyk | 7/28/20 9:08:00 PM |
|---|---|---|

Font: (Default) Times New Roman, 12 pt, Not Italic

| Page 25: [35] Formatted | Slawomir Tulaczyk | 7/28/20 9:08:00 PM |
|---|---|---|

Font: (Default) Times New Roman, 12 pt

| Page 25: [36] Formatted | Slawomir Tulaczyk | 7/28/20 9:08:00 PM |
|---|---|---|

Font: (Default) Times New Roman, 12 pt, Not Italic

| Page 25: [37] Formatted | Slawomir Tulaczyk | 7/28/20 9:08:00 PM |
|---|---|---|

Font: (Default) Times New Roman, 12 pt

| Page 25: [38] Formatted | Slawomir Tulaczyk | 7/28/20 9:08:00 PM |
|---|---|---|

Font: (Default) Times New Roman

| Page 25: [39] Formatted | Slawomir Tulaczyk | 7/28/20 9:08:00 PM |
|---|---|---|

---

## Referee Report (RR1)

**Re-review to Tulaczyk and Foley, August 2020**

The paper is improved from the last submission. In particular, the new table summarizing subglacial conductivities and reflection values is a welcome addition, and I think it will be widely referred to in the field. There are, however, some specific points that are underdeveloped, some of which were not addressed from my previous review. Ultimately, I think the paper has provoked some worthwhile debate, so I hope the comments are viewed as being constructive with the aim of further improving the paper.

**Specific points**

(1) **A clearer presentation of how conductivity impacts on decibel reflectivity values is required.** In delineation of basal water/radiometric analysis, the field generally uses the decibel form of the radar power equation and radar reflection coefficient, $[R]\_dB=20*log10|r|$. Table 1 would therefore be greatly improved if dB columns (or dB values in brackets) were added in. The dB reflectivity values should then be discussed in the context of the radar power equation and related uncertainties (attenuation loss, rough-surface scattering etc) when performing radiometric analysis, as I suggested previously.

A point which highlights why this is essential, is the statement in L332 `*This means that high conductivity subglacial materials can appear significantly brighter than subglacial lakes filled with fresh meltwater'* as this is not true in the dB scale (due to dB reflectivity values being `compressed' for bright reflectors). For example, from Table 1, dB reflection values at 10 MHz are: Clay(10 MHz) =20*log10(0.878) = -1.1 dB and Water(10 MHZ)  =20*log10(0.724) =-2.8 dB. This < 2 dB difference would not be measurable/significant given other uncertainties in the radar power equation, especially since lakes are likely to be a more specular reflector than clay (therefore off-setting the brightness difference).  In my view: `*This means that high conductivity subglacial materials can be of comparable brightness to subglacial lakes filled with fresh meltwater'* is more accurate given inherent uncertainties in radiometric analysis. This still a very useful result and conceivably has lead to a false-positive identification of subglacial lakes and electrically deep water (for me, this is the most important take-home message from the paper)

On a related note, I think Fig. 1D also had the [R]_dB values removed from the previous submission, so it would be helpful to add these back in.

(2) **The establishment of high/low loss limits (via psi) should be made specific to the subglacial materials in Table 1.** The value of the control parameter, psi, is critical to the analysis in the paper. I therefore think that extra columns for psi(10 MHz) and psi(100 MHz) are needed so that the reader can connect the loss-regime of these materials to the general theory in Fig. 1B.  I appreciate this is done in part in the text, but this could be much clearer.

1It also makes sense to point out that psi is equivalent to loss tangent, and also occurs as a control parameter in the permittivity form of the equations in Peters et al. 2005. There are circumstances in radar analysis when the loss tangent is used to discriminate/classify geologic materials (e.g. when assessing losses in a material of unknown permittivity, as often is the case in planetary radar), so makes sense to include these psi values in the look-up table for this reason too.

**(3) The `typical frequency range' in radioglaciolology (1-100 MHz) is an underestimate.** Both in the abstract and throughout the article the authors assert that typical frequency range is 1-100 MHz in radioglaciology with 100 MHz representing a `typical high end'. However - this is simply not the case for airborne systems. For example, the radar system summary table in Winter et al. 2017 lists 4 of the 5 radar systems as being above 100 MHz, with 150 MHz being the most common center frequency.

To address this, I recommend adding a new paragraph in the introduction reviewing the frequency of different radar systems used in radioglaciology, making a clearer distinction between ground-based and airborne systems and their relevant frequency ranges (60-200 MHz being typical for airborne systems). Better distinguishing between these radar-system groups would be helpful for the general discussion, as airborne systems need higher conductivity materials to be relevant to the results in this study (in part, it justifies, why Oswald 2008 and other airborne studies have focused on permittivity).

Note: I appreciate that 100 MHz is still preferable to use in the plots due to use of 1 and 10 MHz.

Winter, A., Steinhage, D., Arnold, E. J., Blankenship, D. D., Cavitte, M. G. P., Corr, H. F. J., Paden, J. D., Urbini, S., Young, D. A., and Eisen, O.: Comparison of measurements from different radio-echo sounding systems and synchronization with the ice core at Dome C, Antarctica, The Cryosphere, 11, 653–668, https://doi.org/10.5194/tc-11-653-2017, 2017.

**(4) Justification for the `wavenumber version' of Fresnel equation**

L53. ` *We believe that the use of complex variables in past studies may have been a barrier to more widespread consideration of the impact of electrical conductivity on radar reflectivity in radioglaciology.*' To me, this line of reasoning does not make sense as a justification for the Stratton/wavenumber formalism. The E-field reflectivity equations used in the paper still contain a complex variable, eq. 7b. It is just that a complex wavenumber is used rather than the complex permittivity (arguably the square-root in the permittivity-form is annoying though!)

In my view the advantages of the Stratton/wavenumber form are:

  a. The wavenumber form enables a cleaner evaluation of the high-conductivity limit for the reflection coefficient, eq. (12). This is algebraically messier to obtain if you start with the permittivity form with the square-root present.
  b. The conductivity is arguably less `submerged' in the wavenumber form (due to being part of the loss tangent in the permittivity form).

**(5) Title**

I still think it is desirable to add a reference to glaciers – e.g. `T*he role of electrical conductivity in radar wave reflection from glacier beds'*.  The current title could apply to analysis of any EM media and the new contribution is the application to glacier beds. Also, within glaciology the title could also apply to satellite radar, which is a very different frequency band.

---

## Referee Report (RR2)

**Review to Tulaczyk and Foley, 2020**

Tulaczyk and Foley (2020) derive the reflection coefficient for the low loss region, the high loss region, and provide the general reflection coefficient that describes the regions of conductivity that are in between. The revised version of the manuscript highlights the impact that conductivity has on the reflection coefficient and notes that for highly conductive materials with low permittivity values, one could obtain reflection coefficients that are even greater than for a pure water subglacial lake. This is a key point for the glaciological community to understand, as one often quickly attributes a brightly reflecting subsurface as subglacial water; however, Tulaczyk and Foley raise the concern that instead of attributing a strong bed echo directly to subglacial water, we should look for additional constraints such as the phase of the returned signal before making such an assertion, because "subglacial sediments can be conductive enough to produce radar reflectivity that is the same, or higher, than reflectivity from an ice-lake interface", even if they have a lower relative permittivity.

In addition to reminding the community of the significance of conductivity in radar reflection, the reminder of the resistive nature of ice and the role that conductivity plays in radar signal attenuation and reflectivity is appreciated since it is an important aspect of the material property that often gets overlooked.

In the revised manuscript,

- The additions to section 2 (Background) including the physical meaning and interpretation of the control parameter was useful.
- The modified Figure 1 with four panels is much cleaner and easy to follow.
- The addition of Table 1, while similar to compilation tables in Peters et al. 2005 and Christianson et al. 2016, was useful for providing a quick reference/overview of different material permittivity values, conductivity values, and reflectivity values at 10 MHz and 100 MHz.
- The additions to Section 5 have made a stronger case to consider the impact of electrical conductivity on radar bed reflectivity. For example, Section 5 discussed approaches to using radar data to constrain both permittivity and electrical conductivity of subglacial materials, and how the inclusion of electrical conductivity in analysis of radar bed reflectivity could be used when mapping of sub-ice geology (e.g., clay content) and fluid salinity on Earth and icy bodies such as Europa.

**Minor Comments**

It would be useful for the radioglaciological community to see Figure 1C and Figure 2B go out to center frequencies greater than 100 MHz. For example, MCoRDS operates from 140 to 230 MHz and the ApRES operates between 200 to 400MHz. In general, it would be interesting to see the

limits of lossless and high-loss conditions for linear radar frequencies up to around 400 MHz since these are frequencies widely used by the radioglaciology community.

The denoted line numbers below correspond to the "tc-2020-9-manuscript-version3.pdf" document.

**Line 45** The authors note that "inferences about sub-ice conditions, e.g., the presence or absence of subglacial water, are drawn from the lateral variations in radar bed reflectivity." In addition to lateral variations, the authors should consider that these inferences of subglacial water could come from temporal variations in reflectivity; for example, a stationary ApRES system that is deployed for a year to monitor the reflectivity changes at a single location.

**Line 53-55.** Is the use of complex variables actually that significant of a barrier? I would consider removing this sentence as it feels more like a conjecture, and it does not add anything to the manuscript.

**Line 59** insert "the" to read as "In general, the mathematical treatment"

**Table 1 (Lines 229-250):** I would double check the table entries; for example, in the second row of the far right column, the real amplitude reflection coefficient, r, should be a positive number as shown in Equation 8.

**Line 288** insert "of" to read as "a system of two equations"

**Line 289** insert "the" to read as "illustrate the limitations"

**Line 289, Line 323, and Line 365:** Consider using "regimes" or "regions" instead of "regimens"

**Line 295 Equation 11.** I would again note to the reader that you are assuming that both of the media have the magnetic permeability of free space, or at least make it clearer in line 265 that from this point going forward the following equations assume that both of the media have the magnetic permeability of free space.

**Line 305 Equation 12.** While I believe $\beta_2 = \alpha_2$ in this case, I would still change the term after $(\alpha_1 - \alpha_2)^2$ to be $\beta_2^2$, not $\alpha_2^2$, in both the numerator and the denominator for consistency and clarity to the reader.

**Line 399-401** The following lines sound a bit awkward and should be revised for brevity and sentence structure:
*"It would be, of course, best to be actually able to use radar observations to constrain both the permittivity and the electrical conductivity of subglacial materials. One piece of observational evidence, the phase shift of the reflected wave, can be used, at least under some circumstances, to independently check if electrical conductivity of the sub-ice material..."*

**Line 403** While it is noted in the main text that Figure 1D is "plotted for the case of 100 MHz linear frequency", I would also highlight this point for Figure 2A.

**Figure 1D and Figure 2A:** I would again note "plotted for the case of 100 MHz linear frequency" in their captions for clarity.

**Line 415**: In addition to super wide bandwidth radars, I would also note that dual-frequency and multifrequency radar systems (such as a combined HF radar and VHF radar) could help with this.

**Line 428-429** Consider "is a frequency dependent measurement." for brevity

**Line 451** Would this technique allow one to better estimate the salinity of subglacial lakes on Earth and icy planetary bodies than current approaches? Also, what are the limitations? For example, scattering effects at different frequencies could also play a role in the change in received power.

**Figure 2B** Consider adding the electrical conductivity symbol for one of the conductivity values (i.e. $\sigma = 4$ S m$^{-1}$) next to the corresponding line, similar to what is done for Figures 2A and 2D where you showed their relative permittivity symbols for several lines (i.e. $\epsilon_r = 85$, etc.), for clarity.

---

## Author Response (AR2)

**UNIVERSITY OF CALIFORNIA, SANTA CRUZ**

Berkeley • Davis • Irvine • Los Angeles • Merced • Riverside • San Diego • San Francisco     Santa Barbara • Santa Cruz

[Figure]

Dr. Slawek Tulaczyk                    (831) 459-5207 (A112 E&MS)
Department of Earth and Planetary Sciences     (831) 459-3074 (fax)
Earth and Marine Sciences Building           stulaczy@ucsc.edu
University of California
Santa Cruz, CA 95064, USA              Santa Cruz, 10/16/2020

Dear Dr. Karlsson,

I am submitting a revised version of the manuscript TC-2020-9 in two formats, one with all changes accepted and one with marked-up changes. I also attach marked-up pdf files with reviewers' comments. In these files, we indicated (in blue type) our point-by-point responses to the comments of the two reviewers. I believe that we have followed all but one of the reviewers' recommendations. The one we are hesitant to follow is concerned with the suggestion that we should include the full version of the radar equation and discuss attenuation and scattering. We believe that this recommendation, if we would follow it, will result in a significant broadening of manuscript scope and in a significant lengthening of the manuscript as both issues (attenuation and scattering) would require treatment in their own right.

There was one recommendation on which the reviewers were split. It concerned the question of including frequencies higher than 100 MHz in our figures. One reviewer recommended it, and the other one specifically said that it is not necessary. We agree with the latter opinion. The figures are showing order-of-magnitude types of differences between 1, 10, and 100 MHz. It would be odd to add any specific higher frequency (like 150, 200, or 400 MHz), representing the current 'upper limit' of radar frequencies. We believe that our figures are general enough in their current form for a reader to grasp the frequency dependence of bed reflectivity. Any reader who is interested in behavior at some specific frequency higher than 100 MHz can use the equations provided in our manuscript to make these calculations.

As a side note, the reviewers are more optimistic than us about 400 MHz being the current upper limit of radars capable of imaging glacier beds. This is the upper limit of the frequency range for BAS ApRES. As part of the Thwaites project that I'm leading, Dr. TJ Young has used this system at WAIS Divide last season and was only able to 'see' through about half of the ice thickness (ca. 1.5 km). And this is the colder part of the ice thickness in which attenuation rates are lower than in the deeper ice layers. 150 to 200 MHz is the more likely upper end of frequency for radars capable of imaging ice sheet beds. And these frequencies are only marginally different than 100 MHz in the context of the physics discussed in this manuscript. Nonetheless, we followed the reviewers' recommendation and listed the upper limit of the radar frequency range given in the text to 400 MHz.

Similarly, the reviewers appear to be pushing us to include scattering discussion because they assume that scattering can be used in distinguishing reflections from ice-over-sediment interfaces and ice-over-water interfaces. Twice in my career, I had the opportunity to see the West Antarctic ice sheet's underside in a borehole video. Both times the base, which normally moves over weak clay-rich sediments, was smooth at length scales that may be causing radar scattering at frequencies similar to 100 MHz (roughness scales of the order of 0.1 to 1.0). We include a short passage to point out that the interface roughness may be rough enough to cause scattering for ice-over-bedrock interfaces but not for ice-over-sediment interfaces. However, anybody who ever walked over a glacially eroded granite surface will recognize that even such sub-ice interface can be relatively smooth on the scale of radar wavelengths.

Thank you for all the work you put into this manuscript and, please, let me know if you require further changes.

Sincerely,

Dr. Slawek Tulaczyk
Professor of Earth Sciences

**Review to Tulaczyk and Foley, 2020**

Tulaczyk and Foley (2020) derive the reflection coefficient for the low loss region, the high loss region, and provide the general reflection coefficient that describes the regions of conductivity that are in between. The revised version of the manuscript highlights the impact that conductivity has on the reflection coefficient and notes that for highly conductive materials with low permittivity values, one could obtain reflection coefficients that are even greater than for a pure water subglacial lake. This is a key point for the glaciological community to understand, as one often quickly attributes a brightly reflecting subsurface as subglacial water; however, Tulaczyk and Foley raise the concern that instead of attributing a strong bed echo directly to subglacial water, we should look for additional constraints such as the phase of the returned signal before making such an assertion, because "subglacial sediments can be conductive enough to produce radar reflectivity that is the same, or higher, than reflectivity from an ice-lake interface", even if they have a lower relative permittivity.

In addition to reminding the community of the significance of conductivity in radar reflection, the reminder of the resistive nature of ice and the role that conductivity plays in radar signal attenuation and reflectivity is appreciated since it is an important aspect of the material property that often gets overlooked.

In the revised manuscript,

- The additions to section 2 (Background) including the physical meaning and interpretation of the control parameter was useful.
- The modified Figure 1 with four panels is much cleaner and easy to follow.
- The addition of Table 1, while similar to compilation tables in Peters et al. 2005 and Christianson et al. 2016, was useful for providing a quick reference/overview of different material permittivity values, conductivity values, and reflectivity values at 10 MHz and 100 MHz.
- The additions to Section 5 have made a stronger case to consider the impact of electrical conductivity on radar bed reflectivity. For example, Section 5 discussed approaches to using radar data to constrain both permittivity and electrical conductivity of subglacial materials, and how the inclusion of electrical conductivity in analysis of radar bed reflectivity could be used when mapping of sub-ice geology (e.g., clay content) and fluid salinity on Earth and icy bodies such as Europa.

**Minor Comments**

It would be useful for the radioglaciological community to see Figure 1C and Figure 2B go out to center frequencies greater than 100 MHz. For example, MCoRDS operates from 140 to 230 MHz and the ApRES operates between 200 to 400MHz. In general, it would be interesting to see the limits of lossless and high-loss conditions for linear radar frequencies up to around 400 MHz since these are frequencies widely used by the radioglaciology community.

The denoted line numbers below correspond to the "tc-2020-9-manuscript-version3.pdf" document.

**Line 45** The authors note that "inferences about sub-ice conditions, e.g., the presence or absence of subglacial water, are drawn from the lateral variations in radar bed reflectivity." In addition to lateral variations, the authors should consider that these inferences of subglacial water could come from temporal variations in reflectivity; for example, a stationary ApRES system that is deployed for a year to monitor the reflectivity changes at a single location.

'temporal' added and a reference to Chu et al. (2016) added as well

**Line 53-55.** Is the use of complex variables actually that significant of a barrier? I would consider removing this sentence as it feels more like a conjecture, and it does not add anything to the manuscript.

Sentence in lines 53-55 removed.

**Line 59** insert "the" to read as "In general, the mathematical treatment"

'the' added, thank you

**Table 1 (Lines 229-250):** I would double check the table entries; for example, in the second row of the far right column, the real amplitude reflection coefficient, r, should be a positive number as shown in Equation 8. As stated in the caption for this table, r in the last column was calculated using equation 11. However, now we list the absolute value |r| to avoid this confusion.

**Line 288** insert "of" to read as "a system of two equations"

'of' added, thank you

**Line 289** insert "the" to read as "illustrate the limitations"

'the' added, thank you

**Line 289, Line 323, and Line 365:** Consider using "regimes" or "regions" instead of "regimens"

all occurrences of 'regimen' or 'regimens' replaced with 'regime' or 'regimes'

**Line 295 Equation 11.** I would again note to the reader that you are assuming that both of the media have the magnetic permeability of free space, or at least make it clearer in line 265 that from this point going forward the following equations assume that both of the media have the magnetic permeability of free space.

The suggested change has been implemented in the sentence on line 265

**Line 305 Equation 12.** While I believe $\beta_2 = \alpha_2$ in this case, I would still change the term after $(\alpha_1 - \alpha_2)^2$ to be $\beta_2^2$, not $\alpha_2^2$, in both the numerator and the denominator for consistency and clarity to the reader. The recommended change has been implemented in Eq. 12. We also added a few sentences and Eq. 13, which is a re-arranged version of Eq. 12 to solve for conductivity.

**Line 399-401** The following lines sound a bit awkward and should be revised for brevity and sentence structure:

*"It would be, of course, best to be actually able to use radar observations to constrain both the permittivity and the electrical conductivity of subglacial materials. One piece of observational evidence, the phase shift of the reflected wave, can be used, at least under some circumstances, to independently check if electrical conductivity of the sub-ice material..."*

We rephrased this passage. Hopefully it works better now.

**Line 403** While it is noted in the main text that Figure 1D is "plotted for the case of 100 MHz linear frequency", I would also highlight this point for Figure 2A.

We modified this sentence as recommended.

**Figure 1D and Figure 2A:** I would again note "plotted for the case of 100 MHz linear frequency" in their captions for clarity.   Added as recommended.

**Line 415**: In addition to super wide bandwidth radars, I would also note that dual-frequency and multifrequency radar systems (such as a combined HF radar and VHF radar) could help with this.
                                          Added as recommended.

**Line 428-429** Consider "is a frequency dependent measurement." for brevity
                                    Changed as recommended.

**Line 451** Would this technique allow one to better estimate the salinity of subglacial lakes on Earth and icy planetary bodies than current approaches? Also, what are the limitations? For example, scattering effects at different frequencies could also play a role in the change in received power.
    We modified slightly the discussion  relevant to subglacial lakes on Earth and icy planetary bodies.

**Figure 2B** Consider adding the electrical conductivity symbol for one of the conductivity values (i.e. $\sigma = 4$ S m$^{-1}$) next to the corresponding line, similar to what is done for Figures 2A and 2D where you showed their relative permittivity symbols for several lines (i.e. $\epsilon_r = 85$, etc.), for clarity.                   symbol for conductivity added as recommended.

Additional commentary on the point raised by the reviewer in reference to Line 451:
As we clearly highlighted in the beginning of the manuscript, its focus is exclusively on the specular reflectivity of ice-bed interfaces. For clarity of presentation, we are hesitant to bring into this discussion the issue of scattering, or any other factor that appears in the full form of the 'radar equation'. This would only open another 'can of worms' and force us to write entire new sections of this manuscript. We can understand that the reviewers may see scattering as an important issue in investigating ice-water interfaces, but getting into this discussion would represent a significant 'scope creep' for this manuscript. We did include a couple of sentences concerning scattering from ice-bedrock versus ice-sediment interfaces in response to a comment from the other reviewer.

**Re-review to Tulaczyk and Foley, August 2020**

The paper is improved from the last submission. In particular, the new table summarizing subglacial conductivities and reflection values is a welcome addition, and I think it will be widely referred to in the field. There are, however, some specific points that are underdeveloped, some of which were not addressed from my previous review. Ultimately, I think the paper has provoked some worthwhile debate, so I hope the comments are viewed as being constructive with the aim of further improving the paper.

**Specific points**

Ad (1) below: We included values of R in Table 1, as recommended.
We are reluctant to enlarge this paper by including the radar equation and attenuation + scattering. We believe this is beyond the scope of this paper.

(1) **A clearer presentation of how conductivity impacts on decibel reflectivity values is required.** In delineation of basal water/radiometric analysis, the field generally uses the decibel form of the radar power equation and radar reflection coefficient, $[R]\_dB=20*log10|r|$. Table 1 would therefore be greatly improved if dB columns (or dB values in brackets) were added in. The dB reflectivity values should then be discussed in the context of the radar power equation and related uncertainties (attenuation loss, rough-surface scattering etc) when performing radiometric analysis, as I suggested previously.

A point which highlights why this is essential, is the statement in L332 `*This means that high conductivity subglacial materials can appear significantly brighter than subglacial lakes filled with fresh meltwater'* as this is not true in the dB scale (due to dB reflectivity values being `compressed' for bright reflectors). For example, from Table 1, dB reflection values at 10 MHz are: Clay(10 MHz) =20*log10(0.878) = -1.1 dB and Water(10 MHZ) =20*log10(0.724) =-2.8 dB. This < 2 dB difference would not be measurable/significant given other uncertainties in the radar power equation, especially since lakes are likely to be a more specular reflector than clay (therefore off-setting the brightness difference). In my view: `*This means that high conductivity subglacial materials can be of comparable brightness to subglacial lakes filled with fresh meltwater'* is more accurate given inherent uncertainties in radiometric analysis. This still a very useful result and conceivably has lead to a false-positive identification of subglacial lakes and electrically deep water (for me, this is the most important take-home message from the paper)

We did add a couple of sentences to point out that scattering may not help distinguish ice-over-sediement from ice-over-water.

On a related note, I think Fig. 1D also had the [R]_dB values removed from the previous submission, so it would be helpful to add these back in.

we added back the dB scale (2) **The establishment of high/low loss limits (via psi) should be made specific to the subglacial materials in Table 1.** The value of the control parameter, psi, is critical to the analysis in the paper. I therefore think that extra columns for psi(10 MHz) and psi(100 MHz) are needed so that the reader can connect the loss-regime of these materials to the general theory in Fig. 1B. I appreciate this is done in part in the text, but this could be much clearer.

we added a column with psi to Table 1

1It also makes sense to point out that psi is equivalent to loss tangent, and also occurs as a control parameter in the permittivity form of the equations in Peters et al. 2005. There are circumstances in radar analysis when the loss tangent is used to discriminate/classify geologic materials (e.g. when assessing losses in a material of unknown permittivity, as often is the case in planetary radar), so makes sense to include these psi values in the look-up table for this reason too.

**(3) The `typical frequency range' in radioglaciolology (1-100 MHz) is an underestimate.** Both in the abstract and throughout the article the authors assert that typical frequency range is 1-100 MHz in radioglaciology with 100 MHz representing a `typical high end'. However - this is simply not the case for airborne systems. For example, the radar system summary table in Winter et al. 2017 lists 4 of the 5 radar systems as being above 100 MHz, with 150 MHz being the most common center frequency.

To address this, I recommend adding a new paragraph in the introduction reviewing the frequency of different radar systems used in radioglaciology, making a clearer distinction between ground-based and airborne systems and their relevant frequency ranges (60-200 MHz being typical for airborne systems). Better distinguishing between these radar-system groups would be helpful for the general discussion, as airborne systems need higher conductivity materials to be relevant to the results in this study (in part, it justifies, why Oswald 2008 and other airborne studies have focused on permittivity). We inserted a sentence in the introduction and referenced Winter et al. 2017. We also changed the range 1-100 MHz to 1-400 MHz throughout the ms.

Note: I appreciate that 100 MHz is still preferable to use in the plots due to use of 1 and 10 MHz.

Winter, A., Steinhage, D., Arnold, E. J., Blankenship, D. D., Cavitte, M. G. P., Corr, H. F. J., Paden, J. D., Urbini, S., Young, D. A., and Eisen, O.: Comparison of measurements from different radio-echo sounding systems and synchronization with the ice core at Dome C, Antarctica, The Cryosphere, 11, 653–668, https://doi.org/10.5194/tc-11-653-2017, 2017.

**(4) Justification for the `wavenumber version' of Fresnel equation**

L53. ` *We believe that the use of complex variables in past studies may have been a barrier to more widespread consideration of the impact of electrical conductivity on radar reflectivity in radioglaciology.*' To me, this line of reasoning does not make sense as a justification for the Stratton/wavenumber formalism. The E-field reflectivity equations used in the paper still contain a complex variable, eq. 7b. It is just that a complex wavenumber is used rather than the complex permittivity (arguably the square-root in the permittivity-form is annoying though!)

In my view the advantages of the Stratton/wavenumber form are:

   a. The wavenumber form enables a cleaner evaluation of the high-conductivity limit for the reflection coefficient, eq. (12). This is algebraically messier to obtain if you start with the permittivity form with the square-root present.
   b. The conductivity is arguably less `submerged' in the wavenumber form (due to being part of the loss tangent in the permittivity form).

We have already deleted the offending sentence starting on Line 53 in response to comments of the other reviewer. Points a and b are the reasons why we turned to Stratton's wavenumber treatment but we see no need to lengthen the manuscript by including this discussion. The manuscript offers usable equations.

**(5) Title**

I still think it is desirable to add a reference to glaciers – e.g. `T*he role of electrical conductivity in radar wave reflection from glacier beds'*.  The current title could apply to analysis of any EM media and the new contribution is the application to glacier beds. Also, within glaciology the title could also apply to satellite radar, which is a very different frequency band.

We modified the title as recommended.

[revised manuscript text omitted]